# Comparison of uridine and N1-methylpseudouridine mRNA platforms in development of an Andes virus vaccine

Ivan V. Kuzmin [1,2], Ruben Soto Acosta[1,2], Layne Pruitt[1], Perry T. Wasdin [3], Kritika Kedarinath[1,2], Keziah R. Hernandez[1,2], Kristyn A. Gonzales[1], Kharighan Hill[1], Nicole G. Weidner[1], Chad Mire [1,2], Taylor B. Engdahl [3], Woohyun J. Moon[4], Vsevolod Popov[1], James E. Crowe Jr [3], Ivelin S. Georgiev [3], Mariano A. Garcia-Blanco[5,6], Robert K. Abbott [1] ✉ & Alexander Bukreyev [1,2,7] ✉

The rodent-borne Andes virus (ANDV) causes a severe disease in humans. We developed an ANDV mRNA vaccine based on the M segment of the viral genome, either with regular uridine (U-mRNA) or N1-methylpseudouridine (m1Ψ-mRNA). Female mice immunized by m1Ψ-mRNA developed slightly greater germinal center (GC) responses than U-mRNA-immunized mice. Single cell RNA and BCR sequencing of the GC B cells revealed similar levels of activation, except an additional cluster of cells exhibiting interferon response in animals vaccinated with U-mRNA but not m1Ψ-mRNA. Similar immunoglobulin class-switching and somatic hypermutations were observed in response to the vaccines. Female Syrian hamsters were immunized via a prime–boost regimen with two doses of each vaccine. The titers of glycoprotein-binding antibodies were greater for U-mRNA construct than for m1Ψ-mRNA construct; however, the titers of ANDV-neutralizing antibodies were similar. Vaccinated animals were challenged with a lethal dose of ANDV, along with a naïve control group. All control animals and two animals vaccinated with a lower dose of m1Ψ-mRNA succumbed to infection whereas other vaccinated animals survived without evidence of virus replication. The data demonstrate the development of a protective vaccine against ANDV and the lack of a substantial effect of m1Ψ modification on immunogenicity and protection in rodents.

Hantaviruses (Family: Hantaviridae, Order: Bunyavirales) are negative-stranded, tripartite RNA viruses infecting fish, reptiles, and mammals[1]. Zoonotic hantaviruses of clinical significance belong to genus *Orthohantavirus*; they circulate in rodents and are transmitted to humans primarily by inhalation of aerosolized rodent excreta[2–4] followed sometimes by human–human transmission. Old World hantaviruses (such as Puumala, Hantaan, Seoul viruses) cause hemorrhagic fever with renal syndrome with case-fatality rates <1–15%, depending

[1]Department of Pathology, University of Texas Medical Branch, Galveston, TX, USA. [2]Galveston National Laboratory, Galveston, TX, USA. [3]Vanderbilt University Medical Center, Vanderbilt Vaccine Center, Nashville, TN, USA. [4]Acuitas Therapeutics, Vancouver, BC, Canada. [5]Department of Biochemistry and Molecular Biology, University of Texas Medical Branch, Galveston, TX, USA. [6]Department of Microbiology, Immunology and Cancer Biology, University of Virginia, Charlottesville, VA, USA. [7]Department of Microbiology and Immunology, University of Texas Medical Branch, Galveston, TX, USA. ✉e-mail: rkabbott@utmb.edu; alexander.bukreyev@UTMB.EDU

on the specific causative agent, whereas New World hantaviruses cause hantavirus pulmonary syndrome with case-fatality rates up to 40%[3]. Among the latter, most human cases are caused in North America by Sin Nombre virus (SNV) and in South America by Andes virus (ANDV)[2,3].

Hantavirus genomes consist of three RNA segments: small (S), medium (M), and large (L). The S segment codes for nucleoprotein (N); the M segment codes for glycoproteins Gn and Gc in a single open reading frame (ORF) of glycoprotein precursor (GPC); the L segment codes for polymerase protein (L)[5,6]. The synthesized GPC is cleaved to Gn and Gc by host signalases at the pentapeptide motif WAASA[7]. Further, the cleaved Gn and Gc form heterodimer Gn/Gc complexes on the virion surface that interact with host cell receptors and trigger receptor-mediated endocytosis and further pH-driven membrane fusion via conformational changes of Gc[8,9]. It was shown that monoclonal antibodies directed to Gn or Gc of ANDV can efficiently neutralize the virus[10]. Orthohantavirus N is conserved, and viruses from various species demonstrate antigenic cross-reactivity based on the N[11,12]. Glycoprotein cross-reactivity that causes cross-neutralization between certain hantaviruses was demonstrated as well and serves as the basis of cross-protection elicited by hantavirus biologics[13–15].

No vaccines or preventive treatments for orthohantavirus infections have been approved by the US Food and Drug Administration or the European Medicines Agency to date. The only vaccines approved at the national level in China and in the Republic of Korea are inactivated whole-virion vaccines which elicit only moderate protection against Hantaan and Seoul viruses, respectively, but not against hantaviruses of other species[16,17]. In preclinical studies, vesicular stomatitis virus-vectored[18–20], human adenovirus type 5-vectored[21] vaccines against ANDV and SNV, and a vaccinia virus-vectored vaccine against Hantaan virus[15] have been reported. The most substantial efforts were focused on the development of DNA vaccines based on the M segment of the viral genome. DNA vaccines targeting Old World hantaviruses, either mono- or polyvalent, have been evaluated in animal models[13,14,22–24] and underwent initial clinical trials[25–27]. These constructs were highly protective but suffered from the common limitation of DNA vaccines which is low immunogenicity and the resulting need to administer them several times at very high doses.

During recent years, substantial advances have been made in the development of mRNA vaccines against various pathogens[28–30], particularly during the COVID-19 pandemic[31–34]. The mRNA platform has multiple advantages: it is rapidly deployable, highly immunogenic, non-infectious, lacks a viral vector or another carrier which could induce undesirable immune responses, and lacks a risk of incorporation into the host's genome[35]. The mRNA vaccine constructs are often packaged in lipid nanoparticles (LNP), which serve multiple purposes: mRNA delivery into cell cytoplasm, protection from host nucleases, and adjuvant effects[35,36].

The immunogenicity of vaccines based on conventional mRNA may actually be reduced due to triggering toll-like receptors (TLRs) 3, 7, and 8, as well as RIG-I receptor resulting in strong induction of the innate immune response[37]. This response leads to the expression and activation of protein kinase R and 2′–5′-oligoadenylate synthase, which in turn leads to a strong suppression of translation of the intended vaccine[38]. In addition, induction of the innate immune response results in the degradation of cellular and ribosomal RNA[38]. Several nucleoside modifications have been designed to combat the induction of the innate immune response by mRNA. Replacement of uridine with pseudouridine (e.g., N1-methylpseudouridine) has been shown to be the most effective[38,39]. It was suggested that cellular pattern recognition receptors (PRR) do not recognize the pseudouridine-containing RNA efficiently, which results in reduced induction of the innate immune response, increased translation of the mRNA vaccine, and improved immunogenicity[28,29,39–41]. Additionally, triggering of the innate immune response occurs through recognition of double-stranded RNA, which is synthesized as a by-product during in-vitro

production of mRNA by the bacteriophage T7 RNA-dependent RNA polymerase. This effect can be greatly reduced by the incorporation of modified nucleosides[42].

On the other hand, non-modified mRNA vaccines also show promise. While the non-modified mRNA COVID-19 HERALD vaccine platform was discontinued by CureVac, it met the prespecified success criteria for efficacy against COVID-19 despite the low 12 µg mRNA dose[43], which is lesser than that of BNT162b2 (30 µg) or Moderna (100 µg) COVID-19 vaccines. Therefore, the unmodified mRNA vaccine platform is also being actively pursued[44,45]. The successful use of some PRR agonists as adjuvants for protein-based vaccines (reviewed in ref. [46]) further supports the argument that induction of innate immune responses by non-modified RNA may be beneficial for the overall vaccine immune response. It is possible that while the innate immune response to non-modified mRNA inhibits the translation, it also stimulates immune cells such as dendritic cells, T cells, and B cells for better adaptive immune response[47,48]. Thus, the net effect of vaccine RNA modification can only be evaluated by a direct comparison of a non-modified and modified mRNA vaccines which otherwise are identical.

In the present study, we focused on the development and evaluation of mRNA vaccines against ANDV because of their public health significance and lack of licensed vaccines. ANDV caused a human outbreak with aggressive airborne human-to-human transmission and multiple fatalities in 2019–2020[49], highlighting its epidemic potential. We generated mRNA vaccines based on non-modified and modified mRNA platforms which were otherwise identical. The two ANDV vaccines were compared for protein expression, innate immunogenicity in vitro, induction of germinal centers (GCs), antibody responses, activation of B cells, B cell receptors (BCRs) repertoires in rodent models, and the protective efficacy in vivo in a 100% lethal rodent model.

## Results

### Design of the ANDV GPC mRNA vaccine construct
Analysis of 70 complete ANDV M segment sequences available in GenBank demonstrated high conservation of the GPC amino acid sequences, with identity values ranging 98.4–100%. The most divergent sequences had only five amino acid substitutions that occurred randomly in the domains corresponding to Gn and Gc. We selected the well-characterized reference sequence NC_003467 for our construct. The 5′ non-coding region of the mRNA originated from the same viral genome that possessed the required minimum Kozak sequence[50]. The ORF (3417 nucleotides including stop-codon) was optimized for expression in human cells using the GenSmart Codon Optimization algorithm available at GenScript, with G + C content of 56.13%, and avoidance of restriction sites used for insertion of the construct into plasmid and its linearization. This ORF was followed with a 3′ non-coding region that consisted of two head-to-tail concatenated sequences of human genomic origin, partial mitochondrially encoded 12S rRNA (mtRNR1) and amino-terminal enhancer of split (AES), a combination that was demonstrated to enhance protein expression from various mRNAs[51]. The 3′ UTR sequence was followed by a poly-A tail of 120 nucleosides, and restriction sites EcoRV and BstBI were separated by a 30-nucleoside poly-A stretch that was used for linearization of the DNA template. The construct (Fig. 1A) was cloned under control of the T7 promoter with an addition of the GGC sequence upstream of the 5′ viral sequence to improve the transcription efficiency.

After linearization of the DNA template with EcoRV and BstBI restriction enzymes, in-vitro transcription was conducted using either the regular uridine (U-mRNA) or N[1]-methylpseudouridine (m1Ψ-mRNA) to generate products with different nucleoside compositions, and cap-1 was added co-transcriptionally. The RNAs were dephosphorylated and purified with cellulose[52].

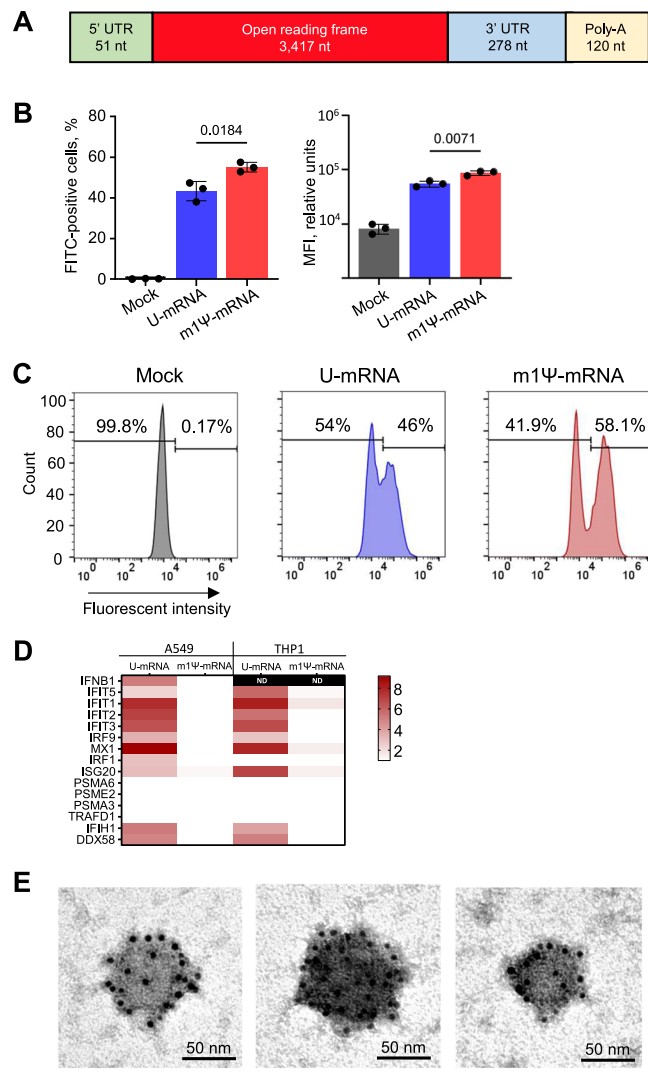

**Fig. 1 | Generation of ANDV mRNA vaccine. A** Schematic representation of the vaccine construct. Expression of Gn/Gc evaluated by flow cytometry in A549 cells at 24 h post transfection with the RNA constructs: mean values from three biologic replicates ± SD, unpaired two-sided *T*-test (**B**) and representative primary data (**C**). **D** Heatmaps of cytokine expression in A549 and THP1 cells at 24 h after transfection with the U-mRNA and m1Ψ-mRNA from biological triplicates, the scale bar shows log2 fold changes; ND not detected; Source data are provided as a Source Data file. **E** Electron microscopy of representative VLPs in supernatants of 293T cells transfected with ANDV U-mRNA. Immunostaining with a primary human ANDV antibody cocktail and secondary 6 nm colloidal gold anti-human antibody; the experiment was repeated twice.

## While the m1Ψ modification of mRNA prevents an innate immune response, its effect on translation is marginal

We evaluated the expression of Gn/Gc in A549 cells at 24 h post transfection with either U-mRNA or m1Ψ-mRNA in equal concentrations by flow cytometry using a cocktail of human monoclonal antibodies that are capable to bind ANDV Gn/Gc antigen and neutralize ANDV in vivo[10], with FITC-labeled secondary antibody. The results demonstrated effective expression of the antigen in cells transfected with either construct (Fig. 1B, C). The proportion of cells expressing Gn/Gc from m1Ψ-mRNA (mean 55.1%; $CI_{95}$ 52.4–57.7%) was greater than from U-mRNA (mean 43.3%; $CI_{95}$ 37.9–48.7%), and the median fluorescence intensity (MFI) of FITC signal from cells transfected with m1Ψ-mRNA (mean 87169.5 reactive units [RU]; $CI_{95}$ 77589.9–96748.7 RU) was statistically greater than the MFI of cells transfected with U-mRNA (mean 54969.33 RU; $CI_{95}$ 47025.1–62913.6). Nevertheless, the overall

difference between the levels of expression from the two vaccine constructs was modest.

To compare the innate immune response triggered by the constructs, A549 and THP1 cells were transfected with either U-mRNA or m1Ψ-mRNA, harvested in 24 h, and mRNA transcripts were evaluated by quantitative RT-PCR (qRT-PCR) using TaqMan™ Array, Human Cytokine Network, pre-defined and custom (Applied Biosystems). In total, the expression of 48 genes was evaluated by ΔΔCt algorithm with 18S or GAPDH as housekeeping genes (Fig. 1D, Table S1). Among type I interferons (IFN-I), only *IFNβ* was detected and could be measured comparatively in A549 but not THP1 cells; type II interferon (*IFNγ*) was not expressed under any treatment; and type III interferons (*IFNλ1, IFNλ2, IFNλ4*) were expressed in the cells transfected with U-mRNA but not in the mock-transfected cells or in the cells transfected with m1Ψ-mRNA. In addition to interferons, 14 IFN-stimulated and other innate immune genes were expressed in the cells transfected with U-mRNA but not in the cells transfected with m1Ψ-mRNA. These data demonstrate that U-mRNA triggers an effective innate immune response, while m1Ψ-mRNA does not, which is consistent with previous reports[37].

In another experiment, we immunized mice intramuscularly with 10 µg of LNP-encapsulated U-mRNA or m1Ψ-mRNA and evaluated cytokine responses in the muscle or the draining popliteal lymph node 24 and 72 h post vaccination by the same approach using a mouse cytokine panel available at Applied Biosystems. We did not observe any detectable differences between cytokine expression levels in the tissues of vaccinated mice compared to those of mice injected with empty LNP (Table S2), which suggests that the innate immune response to vaccination, even if observed in certain cell types, does not necessarily translate into a substantial detectable response in vivo.

## The expressed ANDV Gn/Gc self-assemble into virus-like particles (VLP) or large protein aggregates

Previous studies reported that glycoproteins of bunyaviruses[53] including hantaviruses[54] can self-assemble into virus-like particles (VLPs) in the absence of other viral proteins. We were interested to assess whether ANDV Gn/Gc expressed from our mRNA constructs are capable to form VLPs. Clarified supernatant from 293T cells transfected with ANDV mRNA was concentrated 250× by ultracentrifugation at 100,000 × *g* and evaluated via transmission electron microscopy with uranyl acetate counterstaining. The observed roughly rounded structures were compatible to hantavirus particles, 56–95 nm in diameter (mean 72 nm, $CI_{95}$ 65.6–78.4), and their specificity was further confirmed by immunogold staining (Fig. 1E). However, we did not observe the surface grid-like pattern[55,56] or structures resembling membrane envelope and surface projectiles typical for ectodomain spikes on hantavirus virions[56–59]. Therefore, we cannot completely rule out the possibility that Gn/Gc complexes formed large protein aggregates rather than VLPs. Negative control specimens evaluated by the same method did not show positive immunostaining. It is possible that a formation of VLPs or large glycoprotein aggregates with particulate presentation of the antigens improve immune response, although we only demonstrated these structures in vitro and do not know if the same occurs in vaccinated animals.

## m1Ψ modification of mRNA does not substantially affect the germinal center response in draining lymph nodes

The majority of vaccines work by inducing protective antibody responses[60]. To produce these antibody responses, antigen-activated B cells rapidly proliferate and enter GCs following vaccination[61]. GCs are specialized microstructures in lymph nodes where B cell clones undergo further differentiation[62], somatically mutate their BCRs, and develop affinity-matured protective antibodies in the weeks following vaccination[62]. Therefore, it is critical to evaluate how efficiently novel vaccine constructs can induce GC responses.

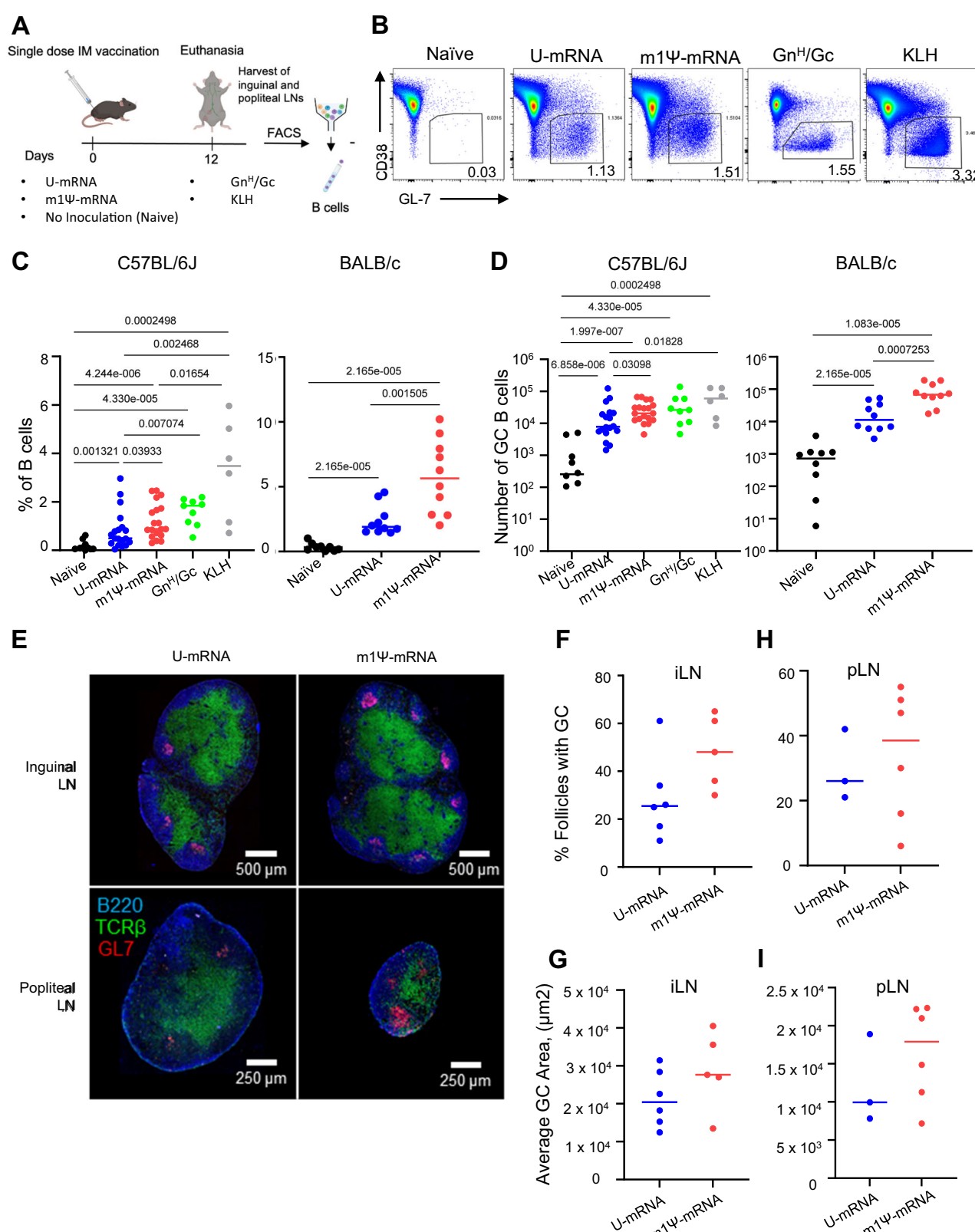

To assess the ability of our mRNA vaccine constructs to induce GC responses, 10–12-week-old C57BL/6J mice or BALB/c mice were immunized intramuscularly with a single dose (5 μg) of either U-mRNA, Ψ-mRNA vaccine (both LNP-encapsulated), or a recombinant Gn^H/Gc antigen[8] (20 μg) mixed with alum (Alhydrogel 2%). Controls included both naïve mice and mice immunized with the irrelevant antigen keyhole limpet hemocyanin (KLH) (20 μg) (Fig. 2A). On day 12 following vaccination we observed a statistically significant induction of GCs in all vaccine groups by flow cytometric analysis. GCs were gated as GL7^+CD38^- among all live B cells (scatter, singlet, live/dead, CD4^-, B220^+) in the draining lymph nodes (Fig. 2B). There was considerable overlap in the GC response to both mRNA vaccines. Overall, the magnitude of this response was comparable between the vaccine groups as measured by the frequency of total B cells (Fig. 2C) or

**Fig. 2 | Single doses of ANDV U-mRNA and Ψ-mRNA induce comparable germinal center and T follicular helper response in vaccinated mice.** **A** Schematic of the studies. **B** Representative flow cytometry plots of GCs from each group of mice. GC B cells were gated as Lymph/Singlet/Live/B220+/CD38−/GL7+. Quantitation of GC B cell frequencies as indicated in (**A**) on day 12 post immunization: proportion (**C**) and absolute counts plotted in log scale (**D**). Data for C57BL/6J mice is pooled from four independent experiments performed under identical conditions (naïve = 10 mice, U-mRNA = 18 mice, m1Ψ-mRNA = 19 mice, GnH/Gc = 9 mice, KLH = 6 mice). Data for BALB/c mice is pooled from two identical experiments (naïve = 10 mice, U-mRNA = 10 mice, m1Ψ-mRNA = 10 mice). **E** Representative histology image of GC B cells from female C57BL/6J mouse immunized with 5 µg dose of indicated vaccines. Blue stain is specific for B220 (B cell marker), green for TCR-β (T cell marker), and red for GL7 (GC B cell marker). **F** Quantification of GC B cells within follicles in inguinal lymph nodes, pooled from two experiments. The total numbers of follicles analyzed for the U-mRNA and m1Ψ-mRNA groups were 597 and 762, respectively. Each dot represents the average GC occupancy for an individual mouse. **G** Quantification of the area of germinal centers in inguinal lymph nodes, pooled from two experiments. The total numbers of GCs analyzed for the U-mRNA and m1Ψ-mRNA groups were 148 and 389, respectively. Each dot represents the average GC area for an individual mouse. **H** Quantification of GC B cells within follicles in popliteal lymph nodes, pooled from two experiments. The total numbers of follicles analyzed for the U-mRNA and m1Ψ-mRNA groups were 266 and 248, respectively. **I** Quantification of the area of germinal centers in popliteal lymph nodes, pooled from two experiments. The total numbers of GCs analyzed for the U-mRNA and m1Ψ-mRNA groups were 72 and 95, respectively (**F−I**). Median values, two-tailed unpaired Mann–Whitney test. Source data are provided as a Source Data file. **A** Created with BioRender.com released under a Creative Commons Attribution-NonCommercial-NoDerivs 4.0 International license.

assessed by absolute number (Fig. 2D) with a tendency for the Ψ-mRNA vaccine to give a slightly greater response. GC B cells were also analyzed histologically in both inguinal and popliteal lymph nodes of C57BL/6J mice immunized with either U-mRNA or Ψ-mRNA vaccines (Fig. 2E). In both lymph nodes, the total relative number of individual GCs, and average GC size were highly similar in both vaccine groups (Fig. 2F–I).

The process of affinity maturation is mechanistically mediated by a cyclic reentry process in which GC B cells shuttle between the dark zone (DZ) and light zone (LZ) of the GC[63–65]. We next assessed if the mRNA vaccines elicit differential composition of DZ and LZ phenotypes within the GCs they induce. Flow cytometric analysis for DZ (live, B220+, CD4−, GL7+, CD38−, CXCR4+, CD86−) or LZ (live, B220+, CD4−, GL7+, CD38−, CXCR4−, CD86+) GC B cells revealed that both vaccines induced comparable DZ and LZ phenotypes (Fig. S1A, B).

We also evaluated how effective each mRNA vaccine is at inducing GC responses at a very low dose of 1.5 µg, which demonstrated rather similar responses (Fig. S2). Furthermore, we evaluated T follicular helper cell (Tfh) responses as they are essential for driving effective GC responses[66]. Both mRNA vaccines elicited rather comparable Tfh responses in both C57BL/6J and BALB/c mice with a tendency for the Ψ-mRNA vaccine to give a slightly greater response (Fig. S2). The antigen-specific IgM and IgG responses were comparable across the vaccine doses and the mouse strains, with IgG being somewhat greater for the Ψ-mRNA as compared to the U-mRNA vaccine (Fig. S3). Taken together, these results suggest both ANDV mRNA vaccine constructs were equally capable of inducing GC B cell responses in vivo.

## m1Ψ modification of mRNA does not significantly affect the transcriptional activation of germinal center B cells in draining lymph nodes

Type I interferons produced by dendritic cells promote their phenotypic and functional activation[67] and also promote expansion and differentiation of T lymphocytes and antibody-producing B cells[68]. As the U-mRNA vaccine but not the m1Ψ-mRNA vaccine induced IFN-I response (Fig. 1B), we were interested to compare the activation of GC B cells in draining lymph nodes after the vaccination. We used BALB/c mice for single-cell sequencing because they mount a more robust antibody response as compared to C57BL/6 mice[69], and the number of isolated GC B cells could be a limiting factor for single-cell sequencing. Groups of 10-week-old BALB/c mice were immunized intramuscularly (IM) with a single 25 µg dose of ANDV mRNA vaccines or a single 20 µg dose of ANDV GnH/Gc-Alum antigen as a protein vaccine control. A separate group of mice were vaccinated with empty LNP to serve as negative controls. Twelve days post vaccination the animals were euthanized, their popliteal and inguinal lymph nodes harvested, GC B cells were purified by flow cytometry and subjected to single-cell RNA-seq (Fig. 3A) except the negative control (empty LNP) where GCs were not observed and unpurified lymphocytes were used at the control

instead. As expected, uniform manifold approximation and projection (UMAP) and principal component analysis (PCA) (Figs. 3B, S4A, B) showed that unpurified lymphocytes from mice that were mock immunized with empty LNP formed a separate group in the UMAP and fell along a distinct axis in the PCA. GC B cells from mice immunized with either U-mRNA, m1Ψ-mRNA, or GnH/Gc overlapped in the UMAP projection, suggesting similar transcriptional profiles. Unsupervised clustering of these cells further supported these observations, as cells from mice mock immunized with empty LNP formed a unique cluster. The rest of the clusters all contained a mixture of cells across the treatment groups, except for cluster 9, which almost exclusively (92%) contained cells from mice immunized with U-mRNA vaccine. Cells in this cluster demonstrated upregulation of genes and enrichment of pathways involved in the innate immune response (Fig. S4C). The upregulation of genes in cluster 9, which was induced only by the U-mRNA vaccine is consistent with the proinflammatory transcriptional profile of this vaccine in transfected A549 and THP1 cells (Fig. 1B). Although statistically significant differentially expressed genes were found between the groups of immunized mice when comparing across cells in all clusters, the differences in expression were subtle between the U-mRNA and m1Ψ-mRNA groups (Fig. S5). These data suggest that the overall transcriptional profile of the GC B cells based on all 11 clusters is similar for the two vaccines. In order to quantify these differences while accounting for differences in total cell counts across groups and mouse repeats, compositional analysis[70] of the clusters across the treatment groups was performed (Fig. 3C). The U-mRNA and m1Ψ-mRNA groups showed comparable composition of cell counts by cluster, with the only quantifiable difference being the presence of Cluster 9 cells for U-mRNA treated mice. Direct comparison of these groups using Hamiltonian Monte Carlo sampling to estimate the expected population of cells in Cluster 9 based on the distribution of cells in the m1Ψ-mRNA group confirmed this difference. Cluster 9 was significantly enriched with a $\log_2$-fold change of $3.99 \pm 0.35$ and an inclusion probability of 1.0 in U-mRNA cells when compared to the m1Ψ-mRNA group (Fig. 3D).

Cells from the three treatment groups showed upregulation of AID (AICDA) and BCL6, with downregulation of IGHD and CCR7 (Fig. 3E), confirming that these groups consist primarily of GC B cells. The negative control empty LNP group cells displayed the inverse expression of the markers, with downregulation of AID and BCL6 and upregulation of IGHD and CCR7, suggesting that these unpurified lymphocytes contain a population of naïve B cells which have not undergone a GC response. Overall, the expression of these markers appears consistent across mice within each group.

Next, the expression of common B cell activation markers was compared across the treatment groups (Fig. 3F). As expected, unpurified lymphocytes from the empty LNP control group showed little or no activation of these markers expressed by B cells. Activation of some of the markers (particularly CD79B) was consistent with

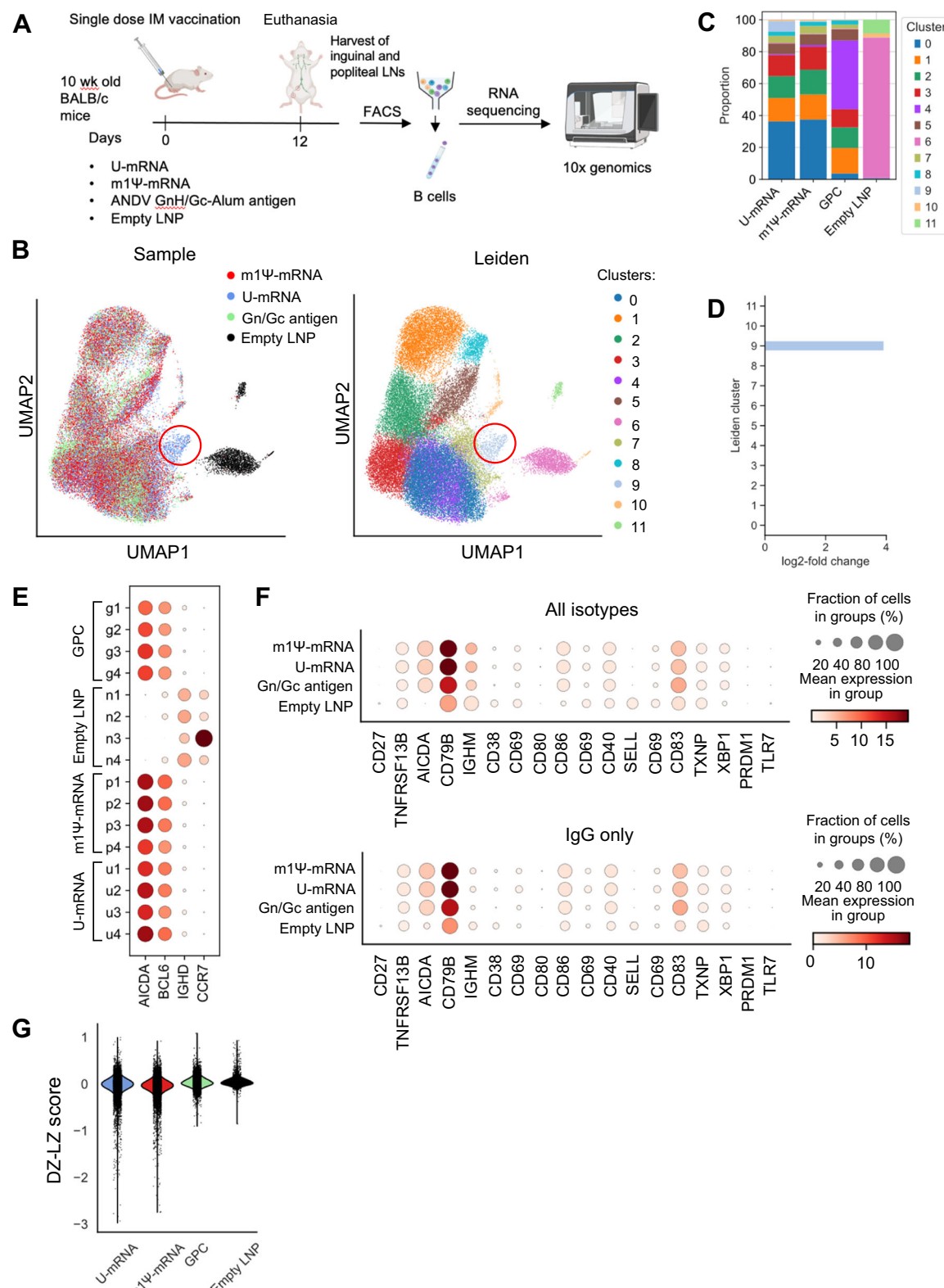

immunostimulatory effects of LNP[36]. Both groups vaccinated with the two mRNA vaccines demonstrated a strong upregulation of multiple markers of activation, particularly CD79B, as well as CD83 and AICDA, and overall showed almost identical expression of the markers. The B cells from mice vaccinated with the Gn^H/Gc protein showed lower mean expression of CD79B and slightly higher mean expression of CD83 compared to the mRNA vaccine groups. In addition to these B cell activation markers, the cells were scored based on the expression

of genes found in LZ and DZ. B cells with more negative scores are DZ-like, and cells with higher, more positive, scores are LZ-like (Fig. 3G). B cells from mice vaccinated with either U-mRNA or m1Ψ-mRNA showed enrichment of cells with lower DZ–LZ scores compared to the mice vaccinated with the Gn^H/Gc protein, suggesting more DZ enrichment. Furthermore, for the mice immunized with empty LNPs, the lympho-cyte scores were primarily close to 0, suggesting very little DZ–LZ characteristics as expected, and DZ–LZ scores did not parallel the

**Fig. 3 | Single doses of ANDV U-mRNA and Ψ-mRNA induce similar gene expression patterns in the lymph node B cells of vaccinated mice. A** Schematic representation of the experiment: BALB/c mice were immunized i.m. with a single 25 μg dose of ANDV mRNA vaccines, with a single 20 μg dose of ANDV GnH/Gc-Alum antigen, or with empty LNP (n = 4 animals/group). Twelve days post vaccination, B cells were isolated from draining lymph nodes and subjected to single-cell RNA-seq and BCR sequencing. **B** Two-dimensional UMAP projection of B cell single-cell RNA-seq profiles across all vaccination groups. Unsupervised clustering of the B cells based on gene expression yielded 12 distinct clusters. Immunoglobulin genes were excluded before clustering. Left: cells colored by treatment group origin, right: cells colored based on cluster assignment with Leiden algorithm. Cluster 9, which was present in mice immunized with U-mRNA only is highlighted with a red circle. **C** scCODA compositional analysis of clusters across treatment groups. The U-mRNA and m1Ψ-mRNA vaccine groups show comparable cluster proportions aside from the presence of Cluster 9 in the U-mRNA group. **D** Quantification of

changes in Cluster 9 cells between the U-mRNA and m1Ψ-mRNA groups. The No U-Turn Sampler (NUTS) model was used in scCODA to estimate the changes in the expected cluster cell populations. For all clusters aside from 9, the predicted cell populations are consistent between the two groups, but the U-mRNA group had a 3.92 $\log_2$ fold change increase in Cluster 9 cells. **E** Expression of germinal center B cell markers by individual mouse within each treatment group. Upregulation of *AID* (*AICDA*) and *BCL6*, with downregulation of *IGHD* and *CCR7* suggest that the three treatment groups consist almost entirely of germinal center B cells, while the negative control B cells are non-germinal center. **F** Mean expression of B cell activation and plasma cell markers across the groups. **G** Distribution of DZ–LZ scores across cells within treatment groups. Cells were scored based on the gene set "GSE38696_LIGHT_ZONE_VS_DARK_ZONE_BCELL_UP", where a higher score is more likely an LZ B cell. Source data are provided as a Source Data file. **A** Created with BioRender.com released under a Creative Commons Attribution-NonCommercial-NoDerivs 4.0 International license.

development of clusters in the GC B cells. Overall, these data demonstrate that despite the absence of the innate immune response to m1Ψ-modified mRNA observed in vitro (Fig. 1D, Table S1), the modification had only marginal effects on the activation of lymph node GC B cells in vivo.

The vaccine mRNA may be detectable in vaccinated mice during several days post-immunization[71,72], and in the GCs of human lymph nodes up to 2 months[73]. We therefore were interested to check if the vaccine mRNA was detectable in the GCs of mouse lymph nodes. The vaccine mRNA was detected in all mice across the two vaccinated groups (Fig. S6). The percentages of cells retaining any amount of the transcripts varied highly across mice within and between vaccine groups, ranging from 0.4% to 5.2% of the total B cells in each mouse although in general this proportion tended to be greater in mice vaccinated with m1Ψ-mRNA. The GC B cells containing the vaccine RNA were uniformly distributed across the UMAP projection of the scRNA-seq data and appeared in small proportions of cells within every cluster including across B cells with all heavy chain isotypes. For example, we did not find a greater proportion of cells containing the vaccine RNA in cluster 9 which exhibited overexpression of innate immune genes.

The persistence of a vaccine antigen may be a beneficial overall aspect of mRNA vaccines, as extended-release regimens have been shown to potentiate GC responses[74,75]. Therefore, we sought to determine whether vaccine antigen expression may be detected in lymph nodes post-immunization. Mice were vaccinated intramuscularly with 10 μg of LNP-encapsulated U-mRNA or m1Ψ-mRNA, and their draining lymph nodes were harvested on days 1, 3, 6, or 14 post injection. The single-cell suspensions were prepared and stained for immune cell markers and for ANDV antigen with further testing by flow cytometry. The population of myeloid cells of immunized mice (which included granulocytes, monocytes, macrophages, and dendritic cells) demonstrated a greater proportion of ANDV-positive cells compared to the background staining level in the control mice (Fig. S7), however, it was as little as <0.003% (approximately <50 events in a population of 1.7–4 × 10⁶ cells), and as such we cannot consider this observation as a reliable evidence of protein expression. No signal suggestive of ANDV antigen was detected in either B or T cells of immunized mice compared to the background documented in the control mice injected with empty LNP. In parallel experiments, GC B cells were also assessed to see if they express ANDV antigen. ANDV antigen was not readily detectable in GC B cells with only sporadic detection above our background control GCs that were induced using an irrelevant antigen (KLH) (Fig. S8). Further detailed studies are needed to determine if a meaningful expression of the vaccine antigen in lymph nodes does occur, and whether the documented persistence of the vaccine mRNA occurred within the GC B cells (as opposed to mRNA inside LNPs that may be surface bound to B cells), and a potential biological significance of that.

## ANDV U-mRNA and m1Ψ-mRNA vaccines induce equal class-switching and somatic hypermutation in draining lymph nodes

As discussed above, IFN-I promotes phenotypic and functional activation of dendritic cells and promotes expansion and differentiation of the adaptive immune response. We, therefore, sought to compare the difference in class-switching and somatic hypermutation (SHM) in GC B cells from draining lymph nodes following vaccination with U-mRNA (high innate immunogenicity) and m1Ψ-mRNA (low innate immunogenicity). To achieve this, we performed single-cell BCR sequencing of RNA isolated from FACS-purified GC B cells recovered from draining lymph nodes on day 12 after vaccination in the mouse study shown in Fig. 3A. As expected, compared to the unpurified lymphocytes from the control group mock-vaccinated with empty LNP, the two mRNA vaccines increased the proportion of GC B cells positive for IGHG1, IGH2B, IGH2C, and IGH3, but reduced the proportion of cells positive for IGHM (Fig. 4A), suggesting induction of a balanced Th1/Th2 response. Similar to the analysis of the B cell transcriptional activation profiles (Fig. 3F), a difference was not detected between the responses induced by the two mRNA vaccines. In contrast, vaccination with GnH/Gc-Alum demonstrated an increased proportion of IGHG1 and only a negligible proportion of IGHG2B, HGHG2C, and IGH3, consistent with induction of a Th2 type response by this adjuvant.

The CDRH3 length did not change across any of the treatment groups (Fig. 4B), and a difference in clonal diversity was not observed. The comparisons of the mean proportion of SHM did not yield a difference between the U-mRNA and m1Ψ-mRNA groups although SHM in both groups of mice was higher than in the unpurified empty LNP group as expected. SHMs in both mRNA vaccinated groups trended to be higher than in the GnH/Gc-Alum vaccinated group, although did not reach statistical significance (Figs. 4C, S9). The clonal expansion of the three vaccinated groups was comparable, with many highly expanded clonal groups (>10 clones) and far more clonal groups with >1 clone compared to the negative control group (Fig. 4D). These clonal groups were assigned based on matching V genes, with ≥70% CDR3 amino acid identity, enabling the identification of "public clones", or highly similar BCRs, between mice from different groups (Fig. 4E). There was an enrichment in the overlap of these highly similar BCRs between the U-mRNA and m1Ψ-mRNA groups (164 shared clonal groups), compared to the overlap of either group with the GnH/Gc-Alum vaccinated group. This overlap is comparable to the number of shared clonal groups between mice within treatment groups as well (Supplementary Data 1), emphasizing that the presence of public clones is relatively rare even for mice receiving the same vaccine construct. The BCRs sampled across groups show differential usage of heavy and light V genes, but the gene usage was diverse, and no single V gene comprised more than 8% of the sampled repertoire (Fig. 4F). Overall, these data demonstrate that even though the m1Ψ modification effectively reduces induction of the innate immune response, the modification

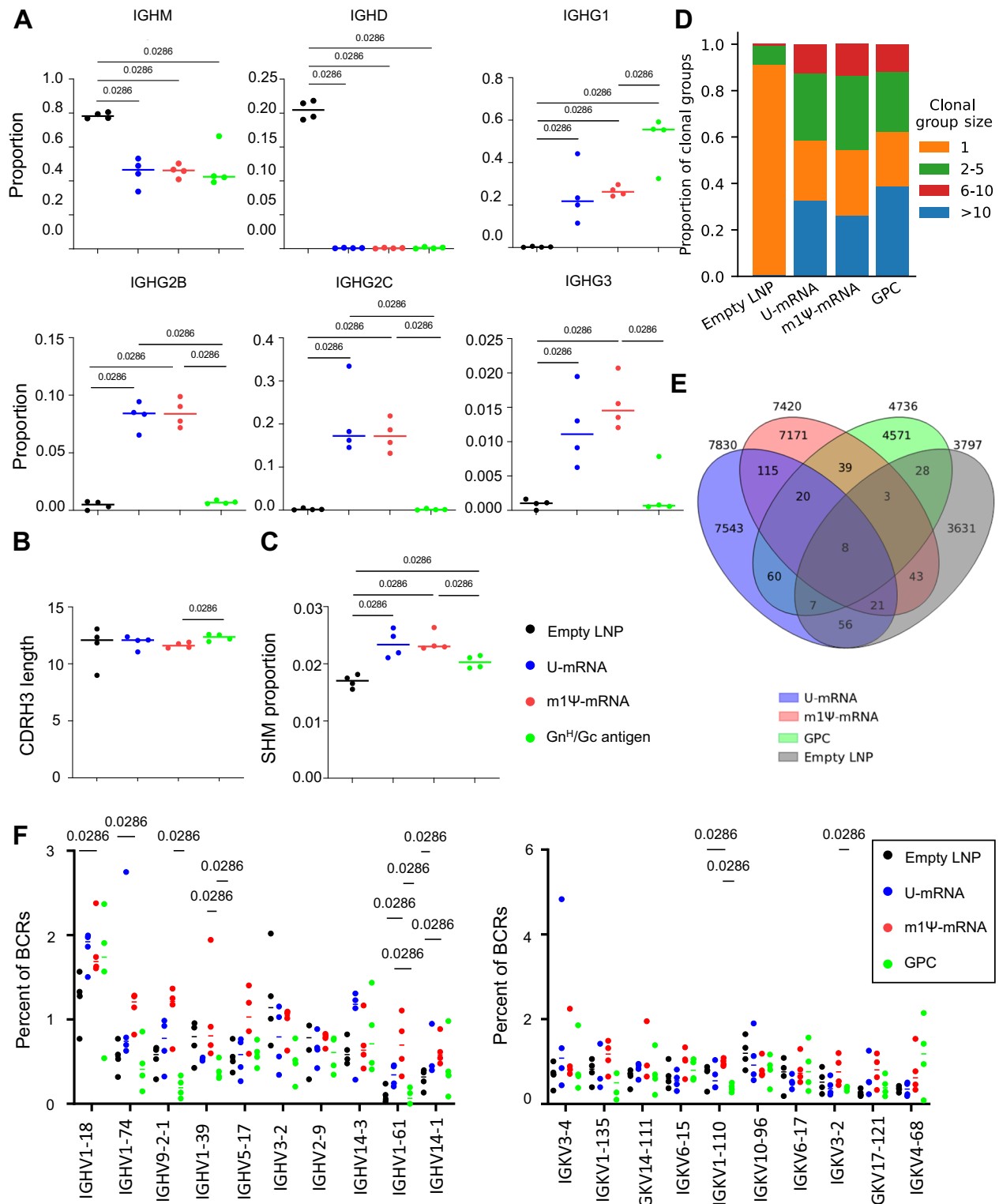

**Fig. 4 | ANDV U-mRNA and m1Ψ-mRNA vaccines induce comparable class-switching and somatic hypermutation in mouse B cells.** B cells from the mouse study presented in Fig. 3 were subjected to single-cell BCR RNA sequencing ($n = 4$ animals/group). **A** Proportions of cells expressing the C region of the heavy chain of various immunoglobulins. **B** Average lengths of antibody heavy chains CDR3 regions (number of amino acids). **C** Average proportion of somatic hypermutation in CDRH3. **D** BCRs were clustered based on Levenshtein distance and matching heavy V genes. Size bins represent a number of BCRs in each clonal family, ranging from unique BCRs to highly expanded clonal groups (>10). **E** Number of overlapping clones between U-mRNA and m1Ψ-mRNA vaccine groups. Little overlap between the groups was observed, but U-mRNA and m1Ψ-mRNA vaccine groups showed the most with seven public clonal groups. **F** U-mRNA and m1Ψ-mRNA vaccine groups show different V gene usage for heavy (left) and light (right) chains. U-mRNA and m1Ψ-mRNA vaccine groups both used IGHV1-18 and IGHV-74 with the highest frequency, but there were differences at lower ranks. Median values, two-tailed unpaired Mann–Whitney test. Source data are provided as a Source Data file.

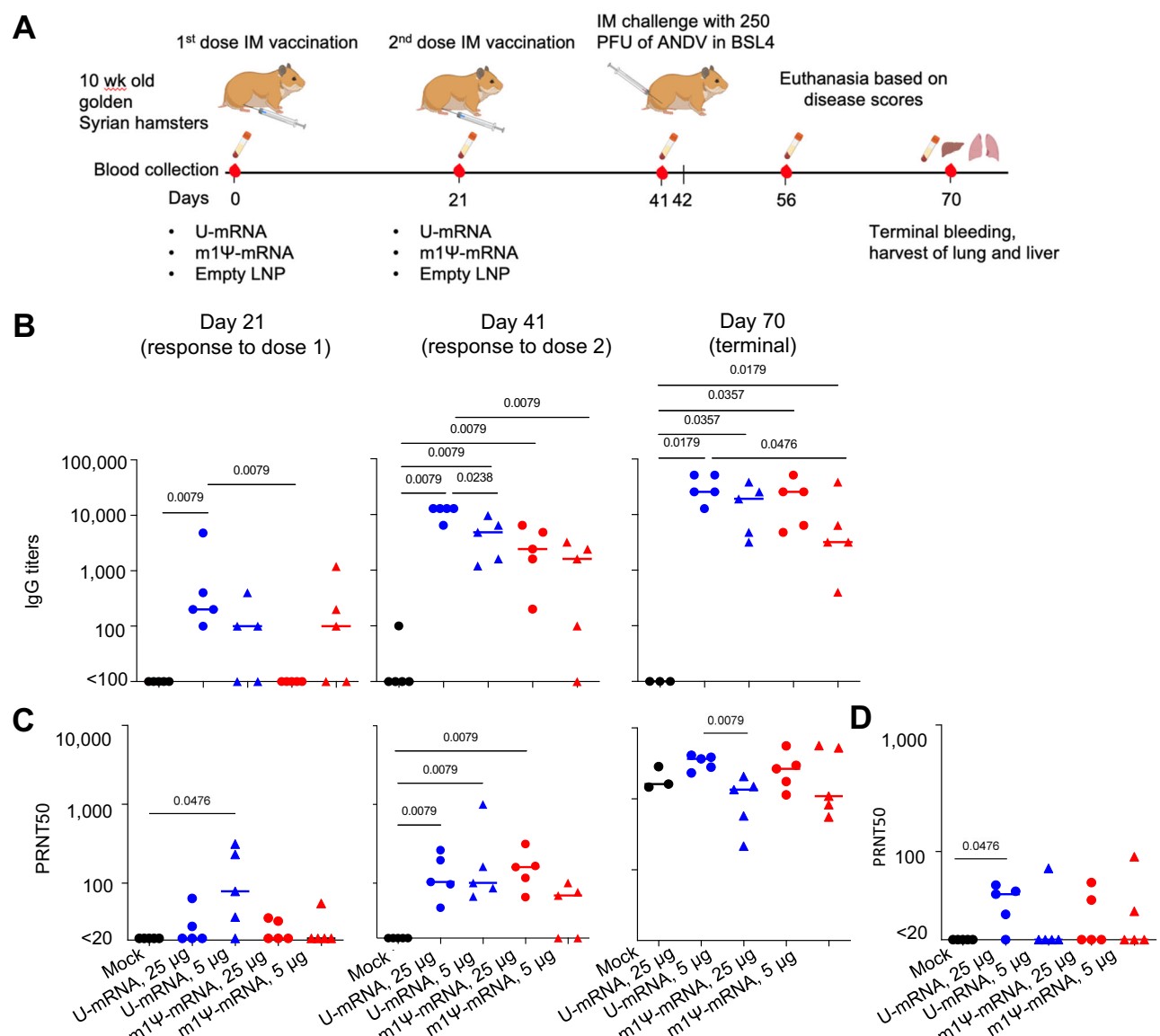

**Fig. 5 | U-mRNA and Ψ-mRNA vaccines induce comparable binding and neutralizing antibody responses against ANDV. A** Schematic representation of the experiment: hamsters were immunized twice with ANDV mRNA vaccines and, along with naïve control, challenged with a lethal dose of ANDV (*n* = 5 animals/group). **B** Binding IgG titers determined by ELISA with ANDV Gn/Gc antigen. **C** ANDV-neutralizing antibody titers in the hamster sera. **D** Antibody neutralizing titers against Sin Nombre virus in hamster sera collected on day 41; optical densities at 450 nm (OD450) of sera diluted 1:100. Median values, two-tailed unpaired Mann–Whitney test. Source data are provided as a Source Data file. **A** Created with BioRender.com released under a Creative Commons Attribution-NonCommercial-NoDerivs 4.0 International license.

does not affect the class-switching, induction of SHMs or clonal expansion in draining lymph node GC B cells.

**Both LNP-encapsulated U-mRNA and m1Ψ-mRNA vaccines elicit potent Gn/Gc-binding and virus-neutralizing antibody responses in golden Syrian hamsters**

Golden Syrian hamsters are very susceptible to ANDV infection, enabling protection studies of candidate vaccines. Hamsters were immunized or mock immunized intramuscularly with 5 or 25 μg of ANDV mRNA constructs encapsulated in LNP on days 0 and 21 (Fig. 5A) and challenged with ANDV on day 42 (21 days after the boost). The animals were observed until day 28 post challenge (dpc). Blood serum was collected at days 0 (baseline), 21 (pre-boost), 41 (pre-challenge), and 70 (terminal, 28 dpc).

Serum IgG binding to ANDV glycoproteins (Gn/Gc-IgG) was determined in ELISA with recombinant Gn^H/Gc antigen[8] starting from serum dilution 1:100. Twenty-one days after the first vaccination, three of five hamsters immunized with 5 μg of either U- or m1Ψ-mRNA developed detectable Gn/Gc-IgG in serum dilutions 1:100–1:1000, as well as all 5 hamsters immunized with 25 μg of U-mRNA. Hamsters immunized with 25 μg of m1Ψ-mRNA vaccine did not have detectable Gn/Gc-IgG at that time. After the boost, Gn/Gc-IgG titers increased in all four groups, except one animal immunized with 5 μg of m1Ψ-mRNA, which showed no detectable antibodies (Fig. 5B).

ANDV-neutralizing antibodies in hamster sera were determined in plaque reduction assay (Fig. 5C). On day 21 after the first vaccination, a fraction of animals in each group showed low but detectable 50% plaque reduction neutralization titers (PRNT50). After the boost, all vaccinated hamsters except two animals vaccinated with 5 μg of m1Ψ-mRNA developed detectable neutralizing antibody titers ranging from 1:49 to 1:946. Although statistically significant differences were not detected between the groups due to the high variability, it is apparent

that animals vaccinated with 5 µg of U-mRNA developed in general greater titers than those vaccinated with 5 µg of m1Ψ-mRNA. In the 25 µg groups, differences were not detected between the U-mRNA and m1Ψ-mRNA constructs. These data suggest that 25 µg of mRNA elicited greater antibody responses than 5 µg of mRNA, whereas differences between U-mRNA and m1Ψ-mRNA constructs were inconsistent. Both vaccines induced potent antibody responses and did not show any benefit of the mRNA modification.

Given previous reports on cross-neutralization between ANDV and SNV[13,14,18], we evaluated hamster sera collected 21 days after the boost for neutralization of SNV (Fig. 5D). A fraction of animals in every group demonstrated SNV-neutralizing activity ($PRNT_{50}$ values of 31–57) with the greater proportion (four of five hamsters) in the group vaccinated with 25 µg of U-mRNA.

### Both LNP-encapsulated U-mRNA and m1Ψ-mRNA vaccines at the higher dose protect equally, whereas U-mRNA appears more efficient than m1Ψ-mRNA at the lower dose

Hamsters were challenged intramuscularly with 250 plaque-forming units (PFU) of ANDV on day 21 after the booster vaccination (Fig. 6A). This dose was uniformly lethal in our previous studies[10,76]. The control animals started showing abrupt disease signs on 9 dpc (lack of movement and reactions to experimenter, hunched posture, rapid shallow breaths) and were euthanized within 24 h with progressive severe respiratory distress. Upon necropsy, large amounts of liquid were found in the hamster thorax, indicating the pulmonary edema. The same occurred with two hamsters vaccinated with 5 µg m1Ψ-mRNA that did not develop neutralizing antibody response to vaccination. Other vaccinated animals survived until the end of observation (28 dpc) without

signs of disease including changes of body weight or temperature (Figs. 6B, C, S19). Liver tissues were collected at necropsy, and high titers of ANDV were detected in all animals that succumbed to the disease. In contrast, virus was not isolated from the liver of any survivor (Fig. 6D).

The infectious challenge further boosted the neutralizing antibody titers. Surprisingly, the mock-vaccinated control group hamsters (based on three samples available for testing) also developed high neutralizing antibody titers after the challenge, although the duration of incubation periods was 9 days only, and the duration of the disease was no longer than 24 h before the animals were euthanized. These serum samples were negative in the Gn/Gc-IgG ELISA (Fig. 5B), suggesting that immunoglobulins of other classes (e.g., IgM) or alternative epitopes not detected by ELISA might mediate virus neutralization. Nevertheless, this virus-neutralization activity failed to protect the non-vaccinated animals from lethal disease.

To further determine if sterilizing immunity was likely achieved, we tested sera of hamsters collected on day 28 after the challenge for antibodies binding to viral nucleoprotein (N) in ELISA. High titers of N-binding antibodies were suggested to be indicative for virus replication upon challenge[23,24]. We used a recombinant truncated N protein based on the sequence of Puumala virus that was shown to cross-react with ANDV N protein[24,77]. The post-challenge data demonstrated no or low levels of N-specific antibodies compared to the positive control that originated from hamsters surviving ANDV infection in our previous experiments (Fig. 6E), suggesting ANDV did not replicate to significant levels in vaccinated animals. Overall, these data demonstrate excellent protective efficacy of the two vaccines at the higher dose and a better antibody response and protective efficacy of U-mRNA over m1Ψ-mRNA at the lower dose.

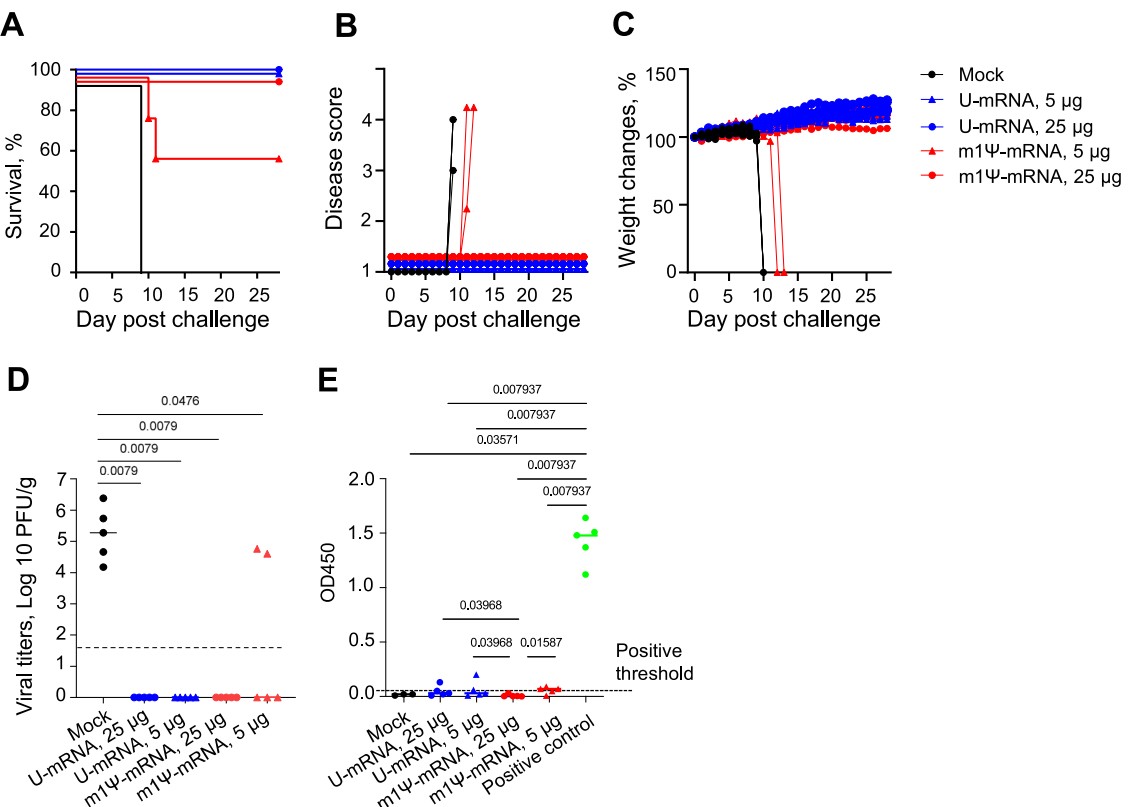

**Fig. 6 | U-mRNA and Ψ-mRNA vaccines protect hamsters from the disease and death caused by ANDV.** The ANDV challenge was performed as indicated in Fig. 5A; *n* = 5 animals/group. **A** The Kaplan–Meier survival curves. **B** The disease scores. Note that the duration of clinical disease was no longer than 24 h before the animals were euthanized. **C** Mean change of weights of animal groups ± SD. **D** Viral load in the liver. The limit of detection is shown by the dashed line. **E** Levels of antibodies binding to Puumala virus N protein in the post-challenge (terminal) sera determined by ELISA; optical densities at 450 nm (OD450) of sera diluted 1:100. Dashed line shows positive threshold at 3× standard deviation of negative sera OD450. Positive control is a set of hamster sera from previous experiments with ANDV infections. **D, E** Median values, two-tailed unpaired Mann–Whitney test. Source data are provided as a Source Data file.

## Discussion

Hantaviruses pose a significant public health threat in parts of the world, and their pandemic potential was highlighted by human-to-human transmission of ANDV in 2019–2020 likely by inhalation of droplets or aerosolized virions[49]. These viruses are prevalent in Asia, and millions of doses of inactivated vaccines against Hantaan and Seoul viruses have been deployed in China and in the Republic of Korea with limited success[16]. In the New World, the prevalence of infection with zoonotic hantaviruses is less, but the severity of human disease is greater, with up to 40% case-fatality rates[3]. It is important to note that the emergence of ANDV and SNV was documented relatively recently, in the 1990s[2], likely due to a combination of increased close contact of humans with rodents and favorable factors for the proliferation of rodent populations in proximity to humans[3,16]. Humankind has learned hard lessons about emerging zoonotic pathogens, including those from the COVID-19 pandemic. There is no guarantee that New World hantaviruses may not cause a similar pandemic in the future, even though at present a large-scale deployment of ANDV or SNV vaccines may not be justified. Furthermore, once we have a successful technology for one of these pathogens, we likely can adapt it to other related viruses, including Old World hantaviruses, as was done with DNA vaccine candidates in the past[11,13,22].

The success of the m1Ψ-mRNA modification led to its incorporation into several mRNA vaccines[31,32,34,78,79]. Additionally, mRNA constructs based on non-modified nucleosides in other studies were capable to elicit robust immune response and protection in various animal models[45,80–82]. However, as mRNA vaccine technology is still relatively new, a direct side-by-side experiential comparison of the two vaccine platforms is required for their comprehensive evaluation. Here we generated two otherwise identical mRNA constructs incorporating uridine (U-mRNA) or $N^1$-methylpseudouridine (m1Ψ-mRNA). We used 5′ non-coding sequence of viral origin as was done previously in hantavirus DNA vaccines[13,23,24,83], although some mRNA studies suggested that a better protein expression may be achieved if eukaryotic (e.g., human) 5′ UTR is used upstream of the protein-coding sequence[44,84,85]. Further studies are needed to evaluate whether the use of human genomic 5′ UTR would improve hantavirus protein expression in human cells. We used a human-derived 3′ UTR previously reported to improve protein expression[51]. Obviously, all these elements are potential subjects of further optimization, along with the length of the poly-A tail, addition of nucleotide spacers, specific residues for folding optimization, and other sequence modifications[28,30,86]. The ANDV Gn and Gc glycoproteins expressed from a single ORF were successfully cleaved in cells and assembled into the typical Gn/Gc structures that were detected by ANDV antibodies. In animal studies, we employed mRNA encapsulated in LNP which not only protect mRNA from fast degradation but also serve as a potent adjuvant[36].

The m1Ψ-mRNA construct demonstrated lack of a significant innate immune response in vitro, in contrast to U-mRNA, which induced strong innate responses (Fig. 1D, Table S1). This observation is in line with previous publications which demonstrated that mRNA composed of modified nucleosides, and particularly of $N^1$-methylpseudouridine, is not recognized by cellular TLRs which results in a reduced type I IFN response, increased translation, and improved adaptive immune response[37,40]. However, despite the high innate immunogenicity of U-mRNA, the level of expression of GPC from this construct was only moderately reduced in transfected cells, as compared to m1Ψ-mRNA (Fig. 1B, C). Furthermore, both constructs induced equal formation of GCs (Fig. 2), comparable class-switching, and SHM in GC B cells in draining lymph nodes (Fig. 4), and similar immunogenicity with U-mRNA being slightly more potent than m1Ψ-mRNA in the Syrian hamster model (Fig. 5). The BCR repertoire for mice treated with the two constructs showed comparable variable gene enrichment and similar clonal expansion, with a higher overlap of public clones in comparison to the control groups (Fig. 4). Comparison of GC B cell activation also demonstrated very similar profiles for the two vaccines, except the presence of the cluster 9 with upregulation of genes and enrichment of pathways involved in the innate immune response in mice immunized with the U-mRNA vaccine only (Figs. 3, S1C).

Golden Syrian hamsters are very susceptible to ANDV infection, enabling protection studies (Figs. 5, 6). The infection causes fast-progressing disease and kills animals within hours after the clinical onset. The viral dose that we used was uniformly lethal to naïve animals in the present and previously published studies[76,87]. In contrast, animals immunized with either vaccine at the higher dose (25 μg) were protected from death and did not show any signs of clinical disease or detectable viral load in the liver. Animals immunized with the lower dose (5 μg) also did not show any signs of clinical disease or detectable viral load in the liver, except for the two animals immunized with m1Ψ-mRNA that did not develop neutralizing antibody responses and developed breakthrough infections. We documented no or barely detectable IgG specific to viral N protein in hamster sera, likely suggesting a lack of replication of the challenge virus.

Serum samples from several ANDV-immunized hamsters cross-neutralized SNV, and the same finding was reported previously for vectored vaccines[18]. We do not know whether the immune response elicited by our ANDV vaccine would protect against SNV upon challenge, as this type of protection cannot be evaluated easily in a conventional rodent model.

Taken together, both murine studies and hamster studies of our U-mRNA and m1Ψ-mRNA vaccine constructs revealed that both vaccines produce comparable humoral immune responses. It is possible that the GC and overall vaccine response may be different between species. However, the serum antibody response was low following a single immunization in both mice and hamsters (Fig. S3 and Fig. 5B), suggesting globally the vaccines are behaving similarly in priming B cells in both species. The nature of the protective antibody response we observed suggests it is likely GC derived. The protective antibody response takes weeks to develop following a boost (Fig. 5B, C) suggesting that the antibody response to both mRNA vaccines is likely generated principally from the GC reaction, as GC-derived serum antibody takes weeks to develop[88]. Alternatively, a limitation is that our recombinant Gn$^H$Gc-based ELISA is not a complete epitope match to the full-length GnGc mRNA vaccines. Therefore, there may be an early antigen-specific antibody response that is immunodominant and simply not detected by our ELISA. Additionally, since GnGc is a relatively complex antigen, degraded "dark antigen" may play a role in the evolution of the antibody response to both mRNA vaccines[89]. One limitation of the study was that we evaluated GC responses only following the priming dose. Since GCs can last many months[90], it would be interesting to assess how each of these mRNA vaccine platforms drive GC durability, and SHMs longitudinally including the following boost.

Why did the m1Ψ-U modification have only a relatively modest effect on the protein expression (Fig. 1C, D), very limited or no effect on the immunogenicity (Figs. 2–5) and no effect on the protective efficacy (Fig. 6) despite the lack of the innate immunogenicity? Several explanations can be suggested. First, the extent of IFN-I-mediated activation of protein kinase R and 2′-5′-oligoadenylate synthetase (which inhibit the translation) varies broadly, and possibly is not very significant in A549 cells used to compare the expression in this study (Fig. 1B, C) and in immune cells involved in the presentation of the antigen in vivo, such as B cells (Fig. 4) and dendritic cells. Second, the relevant cell types may have additional mechanisms which minimize the inhibitory effects of the innate response on the translation. Third, it is also possible that a reduced expression of antigen from non-modified mRNA is compensated by the immunostimulatory effects of the cytokines induced by non-modified but not by modified form of mRNA. Furthermore, IL-1 and IL-1 receptor antagonist, which are key

regulators of the inflammatory response to RNA vaccines, are induced to very different levels in human and rodent cells[91]. In addition, the biological effects of oligonucleotide synthetase 1, whose activation and its subsequent inhibition of translation is prevented by utilization of modified nucleosides[28] is also different in human and mouse cells[92]. Finally, we did not attempt to characterize and compare the CD8 T cell responses to the two vaccines. However, we did observe comparable CD4 Tfh responses between both vaccines in mice (Fig. S2). In conclusion, the demonstration of high immunogenicity and protective efficacy of non-modified mRNA vaccines in non-human primates[44,45], tolerability in human trials[43], and the lack of ribosomal frameshifting demonstrated for N1-methylpseudoudouridine-modified mRNA[93] indicates the high potential of this vaccine platform.

## Methods

The study was performed with all relevant ethical regulations laid out by the University of Texas Medical Branch (UTMB) Institutional Biosafety Committee (IBC) and Institutional Animal Care and Use Committee (IACUC). Animal studies were performed under IACUC protocol 2108048; sex of animals was not under consideration in this study. Animals were housed at the normal 12/12 dark/light cycle, ambient temperature of 18–22 °C, and humidity of 40–50%. Food and water were supplied ad libitum.

### Viruses used in the study

ANDV, isolate Chile-9717869 also referred to as ANDV 9717869 (GenBank Accession Number AF291703, NC_003467), recovered in 1997 from a long-tailed pygmy rice rat (*Oligoryzomys longicaudatus*), was used after eight passages in Vero E6 cells. SNV, isolate SN 77734 (GenBank Accession Number AF281850), recovered in 2006 from a deer mouse (*Peromyscus maniculatus*), NM, USA, was used after two passages in deer mouse and four passages in Vero E6 cells (ATCC #CRL-1586).

### Production of DNA template for ANDV mRNA

The nucleotide sequence of 5′ untranslated region and ORF was obtained from the reference ANDV M segment sequence, GenBank record NC_003467. The ORF was codon-optimized using GenSmart Codon Optimization, Version Beta 1.0 available at GenScript website (https://www.genscript.com/tools/gensmart-codon-optimization; last time accessed on February 15, 2022). Partial mtRNR1 (mitochondrially encoded 12S rRNA) and AES sequences concatenated head-to-tail, used for 3′ untranslated region, were described elsewhere[51]. Poly-A tail of 120 nucleotides was placed downstream, and restriction sites NotI, EcoRV, and BstBI were incorporated for insertion of the construct into plasmid and further linearization. The sequence was placed under T7 promotor with the addition of GGC codon to improve transcription efficiency. The construct was generated by GenScript and supplied in pUC-19 vector.

Plasmid amplification was performed in NEB Stable competent cells (New England Biolabs), and selected clones were sequenced prior to use. The template purified with ZymoPURE Plasmid Maxiprep kit (Zymo Research) was linearized using EcoRV and BstBI restriction enzymes (New England Biolabs).

### mRNA generation, purification, and encapsulation

Two mRNA constructs were prepared from the same DNA template: one with regular uridine (U-mRNA) and another with N$^1$-methylpseudouridine (m1Ψ-mRNA). We followed the protocol published previously[94]. The U-mRNA was generated using HiScribe In-vitro Transcription Kit (New England Biolabs) with CleanCap Reagent AG (TriLink) to obtain cap-1 structure. The m1Ψ-mRNA for in-vitro studies was generated per the same protocol with the only difference that uridine was replaced with N$^1$-methylpseudouridine (TriLink). The m1Ψ-mRNA for in vivo study in hamsters was generated using HighYield T7

Cap 1 AG (3′-OMe) mRNA Synthesis Kit (me1Ψ-UTP) (Jena Bioscience GmbH). The generated RNA was purified with lithium chloride[95] and dephosphorylated with recombinant shrimp alkaline phosphatase (New England Biolabs). Further mRNA was purified from dsRNA contaminants by cellulose as described previously[52], and finally precipitated with sodium acetate[95]. DNA template and RNA concentrations were determined using NanoDrop instrument (Thermo Fisher Scientific). The purified RNA was either used in in-vitro studies or encapsulated into LNP of proprietary formulation by Acuitas Therapeutics (Vancouver, BC, Canada) for subsequent in vivo experiments.

On different steps of preparation, RNA integrity was assessed by electrophoresis on 2% agarose (E-Gel EX, Invitrogen), whereas the presence of dsRNA contaminants prior and after cellulose purification was assessed with J2 monoclonal antibody (Millipore Sigma) via dot-blot on BrightStar Plus positively charged nylon membrane (Invitrogen), further incubated with a secondary goat anti-mouse IgG antibody conjugated with horse radish peroxidase (HRP) (SeraCare), developed using Pierce ECL 2 Western Blotting Substrate (Thermo Fisher Scientific), and detected on Odyssey® XF Imaging System (LI-COR).

### Transfection of cells with ANDV mRNA

A549 cells (ATCC # CCL-185) or 293T cells (ATCC # CRL-3216) grown in 24-well plates during 24 h for 70–90% confluence were transfected with 0.5–1.0 µg of cellulose-purified U-mRNA or m1Ψ-mRNA, using TransIT-mRNA Transfection Kit (Mirus Bio) according to the manufacturer's instructions. THP1 cells (200,000/replicate) were transfected using Lipofectamine Messenger Max (Thermo Fisher Scientific) according to the manufacturer's instructions. Twenty-four hours after transfection, supernatants were removed, cells were washed twice with phosphate-buffered saline (PBS) and collected either for flow cytometry or RNA extraction. All experiments were performed in biological triplicates.

### Flow cytometry of cells transfected with ANDV constructs

Adherent cells transfected with ANDV U-mRNA or m1Ψ-mRNA in 24-well plate format as described above were digested in 0.25 ml of 0.25% trypsin–EDTA (Gibco) which was further neutralized by an addition of 0.25 mL of calf serum (Corning). Cells were washed by PBS, fixed in 4% paraformaldehyde (Polysciences, Inc.), and subjected to immunostaining with a cocktail of human monoclonal antibodies ANDV-4, -5, -12, -22, -23, -34, and -44 targeting ANDV Gn/Gc and capable to both bind and neutralize the virus[10] diluted to 2 µg/mL in StartingBlock T 20 (TBS) Blocking Buffer (Thermo Fisher Scientific). After incubation during 1 h at room temperature and two PBS washes, cells were incubated with the secondary goat anti-human FITC-conjugated antibody (SouthernBiotech) diluted 1:500 in TBS and incubated at the same conditions in the dark. Stained cells were washed twice in PBS and subjected to flow cytometry using an Accuri C6 instrument (BD Biosciences). The results were evaluated with FlowJo software, version 10.8.

### qRT-PCR for cytokine expression

A549 and THP1 cells transfected with ANDV U-mRNA or m1Ψ-mRNA in 24-well plates as described above were digested in 1.0 mL of TRIZol reagent (Invitrogen). Gastrocnemius muscle and draining lymph node of mice vaccinated with 10 µg of either U-mRNA or m1Ψ-mRNA (as described below) were homogenized, and 100 µL of 10% suspension of the homogenate digested in 1.0 mL of TRIZol reagent. After chloroform separation, the aqueous phase was mixed with an equal volume of 100% ethanol and processed with Direct-zol RNA MicroPrep (Zymo Research) with on-column DNAse I treatment. The extracted RNA was quantified using NanoDrop and reverse transcribed with High-Capacity cDNA Reverse Transcription Kit with RNase Inhibitor (Applied Biosystems) according to the manufacturer's instructions. The resulting cDNA was diluted to 4–5 ng/µL, and PCR amplified in the plates of TaqMan™ Array, Human Cytokine Network, Fast 96-well, or

custom arrays for mouse cytokines in the same Fast 96-well format (Applied Biosystems; Cat No 4331182) using TaqMan® Fast Advanced Master Mix (Applied Biosystems), on QuantStudio™ 6 Flex Real-Time PCR System (Applied Biosystems). Results of qRT-PCR were normalized on the expression values of GAPDH or 18S housekeeping genes, and quantified by the ΔΔCt method[96].

### Electron microscopy

293T cells were transfected with 5 μg of ANDV U-mRNA in six-well plates as described above. Forty-eight hours post transfection, supernatant was collected into conical tubes, clarified by centrifugation at $12 \times g$ for 10 min, filtered through Millex®-HP 0.45 μm syringe filter (Millipore Sigma) and concentrated 250× by ultracentrifugation at $100,000 \times g$ for 2 h. Supernatant from 293T cells with added transfection reagents but without mRNA was used as negative control. The samples were incubated on copper grids, counterstained with 2% aqueous uranyl acetate, and observed at JEM-1400 electron microscope (JEOL Ltd.), accelerating voltage 80 kV. To confirm specificity of the observed structures, samples were absorbed to nickel grids and immunostained with primary anti-ANDV human monoclonal antibodies (described above) and secondary 6 nm colloidal gold AffiniPure goat anti-human IgG (H + L) antibody (Jackson ImmunoResearch Laboratories, Inc.) following post-staining fixation in 2% glutaraldehyde and counterstaining with 2% aqueous uranyl acetate.

### Evaluation of innate immune responses in muscles and draining lymph nodes of immunized mice

BALB/c mice, 8-week-old females (The Jackson Laboratory), five animals per group, were vaccinated once intro gastrocnemius muscle with 10 μg of LNP-encapsulated ANDV U-mRNA or m1Ψ-mRNA in 50 μL volume. Empty LNP diluted in PBS at approximately the same concentration were used as a negative control. Twenty-four or 72 h post vaccination the animals were euthanized, the gastrocnemius muscle and draining popliteal lymph node collected, and subjected to cytokine expression analysis as described above.

### Analysis of germinal centers in lymph nodes of immunized mice

For analysis of GC phenotypes in mice, 10–12-week-old female C57BL/6J or BALB/c mice (The Jackson Laboratory) were immunized intramuscularly with one dose (5 μg or 1.5 μg) of either U-mRNA or m1Ψ-mRNA formulated with LNP. Positive control constituted 20 μg of recombinant Gn$^H$/Gc antigen formulated with Alhydrogel (2%) in a 1:1 ratio (total volume 50 μL). Negative control constituted one dose of 20 μg of KLH adjuvant in Alhydrogen (2%) (InvivoGen) in a 1:1 ratio (total volume 50 μL). Inguinal and popliteal lymph nodes were pooled for each mouse on the indicated day. Cells were counted, Fc blocked, and stained with indicated monoclonal antibodies that were commercially sourced (Biolegend and BD Biosciences). Key antibody clones used were: 11-26c.a2 (IgD), 281-2 (CD138), RA3-6B2 (B220), GL7 (GL7), 90 (CD38), GL-1 (CD86), 1.27F12 (CXCR4), GK1.5 (CD4), 53-6.7 (CD8), LI38D7 (CXCR5), 29F.1A12 (PD-1), and MF-14 (FoxP3). The following populations were gated as: GC B cells (scatter/singlet/live/B220+/CD4−/CD38−/GL7+), DZ GC B cells (scatter/singlet/live/B220+/CD4−/CD38−/GL7+/CXCR4+/CD86−), LZ GC B cells (scatter/singlet/live/B220+/CD4−/CD38−/GL7+/CXCR4−/CD86+), and Tfh (scatter/singlet/live/B220−/CD4+/FoxP3−/CXCR5+/PD-1+). Cells were acquired on a BD LSR Fortessa or Agilent NovoCyte Penteon and data were analyzed using Flowjo v10.

Immunohistology analysis was done as follows. Cryosections 5–8 μm thick were cut from inguinal and popliteal lymph nodes frozen in Tissue-Tek OCT compound (Electron Microscopy Sciences) and fixed in a 1:1 mixture of acetone and methanol for 10 min at −30 °C. At the time of staining the sections were rehydrated and blocked in 0.5% bovine serum albumin (BSA) and 0.1% Tween-20/PBS (stain/wash buffer used in all subsequent steps). The slides were treated with a 1:50

dilution of Fc block (clone 2.4G2, BD Biosciences) and 1:10 dilution of naïve C57BL/6J serum. Sections were stained with 1:400 dilutions of the following mAbs: RA3-6B2 (B220), GL7 (GL7), and H57-597 (TCR-β). Coverslips were mounted with Fluoromount-G (SouthernBiotech) prior to image acquisition with a Zeiss Axioscan 7 equipped with Collibri 7 using Zen Blue v 3.7.97.05000. Images were analyzed manually using QuPath v 0.5.1[97]. Detailed information on antibodies used in the study is present in Table S3.

### Analysis of ANDV antigen persistence in draining lymph nodes of vaccinated mice

Groups of five BALB/cJ mice, 10–12 weeks old (The Jackson Laboratory), were vaccinated intramuscularly with one dose (10 μg) of LNP-formulated U-mRNA or m1Ψ-mRNA, or with empty LNP (in approximately the same concentration as in vaccines) as a negative control. Draining inguinal and popliteal lymph nodes were harvested and pooled for each mouse on days 1, 3, 6, and 14 after vaccination. Single cells were isolated by passing via 100 μm cell strainer (Falcon), washed in PBS, and stained with live/dead Zombie Yellow dye (Biolegend), and Fc blocked. This was followed by extracellular staining with indicated monoclonal antibodies that were commercially sourced (Biolegend). Key antibody clones used were: RB6-8C5 (Gr1), M1/70 (CD11b), RA3-6B2 (B220), 17A2 (CD3), GL7 (GL7), 90 (CD38), GL-1 (CD86). Cells were fixed with 4% paraformaldehyde and permeabilized using BD Cytofix/Cytoperm™ Fixation/Permeabilization Kit (BD Biosciences) per manufacturer's instructions. Then, cells underwent intracellular staining using human monoclonal antibody ANDV-4 labeled with Mix-n-Stain™ CF™ 647 Antibody Labeling Kit (Millipore Sigma). Flow cytometry was performed with FACSymphony A5 SE (BD Biosciences) and data were analyzed using FlowJo v10.9.0.

ANDV-positive cells were gated on the following populations: scatter/singlet/live (live cell population), scatter/singlet/live/B220+ (B cells), and scatter/singlet/live/CD3+ (T cells). Additionally, ANDV-positive cells were gated on the following myeloid cell populations: scatter/singlet/live/B220−CD3−/CD11b+ (granulocytes, monocytes/macrophages, dendritic cells), scatter/singlet/live/B220−CD3−/Gr1+ (granulocytes and macrophages), scatter/singlet/live/B220−CD3−/CD11b+/CD86+ (macrophages and dendritic cells), and scatter/singlet/live/B220−CD3−/CD11b+/Gr1+.

### Preparation of B cells from lymph nodes of immunized mice for single-cell sequencing

Ten-week-old female BALB/c mice (The Jackson Laboratory), four per group, were immunized intramuscularly with one dose of 25 μg of either U-mRNA or m1Ψ-mRNA formulated with LNP. Positive control constituted 20 μg of recombinant Gn$^H$/Gc antigen formulated with Alhydrogel (2%) in a 1:1 ratio (total volume 50 μL), and negative control constituted a suspension of empty LNP in approximately the same concentration that was used in mRNA vaccines. In this experiment, lymph node drainage of the vaccine was noted bilaterally. Twelve days after immunization animals were euthanized, and draining inguinal and popliteal lymph nodes were harvested and pooled for each mouse for cell sorting.

GC B cells were gated as single live lymphocytes that are B220+, CD4−, CD8−, GL7+, CD38−, IgD−, CD138− and purified by cell sorting using the BD FACSAria Fusion sorter (BD Biosciences). For negative control, sorting was not performed due to the limited amount of GC B cells induced by the empty LNP vaccination. From this group, total lymphocytes were taken for sequencing.

B cell suspensions were processed for single-cell sequencing following Chromium Next GEM Single Cell 5′ Version 2 protocol. Members of a pool of 750,000 barcodes were sampled individually to index the transcriptome of each cell. This procedure was performed by partitioning each cell into Gel beads-in-emulsion (GEMs) combined with a Master Mix containing reverse transcription (RT) reagents and

poly(dT) RT primers. The emulsion was made using the Chromium Controller device and Next GEM Chips (10x Genomics). The GEM generation and further RT reaction produced 10× barcoded full-length cDNA from poly-adenylated mRNA. This initial cDNA was PCR ampli-fied to generate enough material for 5′ Gene expression and BCR sequencing. Right after PCR amplification, bioanalyzer quality control was performed for all the samples using the Agilent Bioanalyzer High Sensitivity DNA assay in the 2100 expert software (Agilent). All the samples passed the initial QC with a cDNA size of 700–1500 bp.

Amplified full-length cDNA from poly-adenylated mRNA was used to generate 5′ gene expression (GEX) and V(D)J libraries (BCR). For GEX library construction, the cDNA was enzymatically fragmented, and size was selected to optimize the cDNA amplicon size. For V(D)J library construction, full-length cDNA was used to amplify 10x barcoded V(D)J segments by PCR amplification using specific primers for BCR constant regions. BCR transcripts were fragmented and selected to get variable-length fragments that span the V(D)J segments. P5, P7, i5, and i7 sample indexes and Illumina R2 sequence (read 2 primer sequence) were added via End Repair, A-tailing, Adapter ligation and sample index PCR for both GEX and BCR libraries. A second quality control was per-formed for each type of library before sequencing. The electro-pherograms showed a library size of 500–900 bp, the size expected for the GEX constructs, and 600 bp for BCR libraries. Finally, libraries were pooled and sequenced by the New York Genome Center using a NovaSeq sequencer and S2 and SP flow-cell for GEX and BCR libraries respectively. The details on single-cell sequencing are included in Supplementary Data 2.

## Processing of single-cell sequencing data

The output FASTA files from single-cell GEX RNA-sequencing (scRNA-seq) and single-cell BCR-sequencing (scBCR-seq) were processed using Cell Ranger 7.1 (10X Genomics) as described elsewhere[98]. The Cell Ranger *counts* pipeline was used for scRNA-seq data, and the Cell Ranger *vdj* pipeline was used to process scBCR-seq data. For the scRNA-seq pipeline, the vaccine RNA sequence was added to the GRCm38 mouse reference genome prior to alignment. The aligned sequences for gene expression and BCR were then processed in Python using Scanpy 1.9.1[99] and custom scripts. Only barcodes for cells detected in both the scRNA-seq and scBCR-seq data were retained for downstream analyses. Details on the number of cells recovered and metrics from quality control and preprocessing are included in Sup-plementary Data 2.

## scRNA-seq pipeline

**Preprocessing.** Following the alignment of the mRNA reads to the reference genome, the scRNA-seq data were processed in Scanpy. Cells with <5000 total gene counts or a total percentage of reads from mitochondrial genes >5% were removed in order to filter out poor quality and lysed cells. Reads were log-normalized with a scale factor of $10^4$ within each sample and then scaled to unit variance and zero mean. Genes encoding for V(D)J regions of the IGH, IGK, and IGL loci were removed, and then the gene expression matrices from all samples were concatenated prior to clustering. Highly variable genes with a mini-mum dispersion of 0.5 were selected for use in dimensionality reduction and clustering.

**Dimensionality reduction and clustering.** Dimensionality reduction and clustering were performed based on the recommended clustering pipeline provided in the Scanpy documentation. PCA was used to reduce the gene expression matrix to the first 30 principal compo-nents, which were then used to construct a nearest neighbor graph with $k = 20$. UMAP was used to obtain a two-dimensional representa-tion of the data for downstream analyses and visualization, followed by unsupervised clustering using the Scanpy implementation of the Lei-den algorithm[100] with a resolution of 0.5.

**Compositional analysis.** Following cluster assignment for the scRNA-seq analysis in Fig. 3, the Python package scCODA[70] was used to quantify the compositional changes in cell types across treatment groups. The uridine and pseudouridine treatment groups were com-pared directly using the No U-turn sampling method with the largest shared cell type, Cluster 0, as the reference. A false discovery rate of 0.05 was used.

**Further gene expression analysis.** Only IgG-expressing B cells were retained for further analysis, as these B cells are expected to be elicited by the vaccine treatments, and differences in gene expression between B cell subtypes could convolute the downstream analyses. Differential gene expression (DGE) analysis was performed to identify differentially expressed genes between the cells from mice in the U-mRNA and m1Ψ-mRNA groups based on Benjamini–Hochberg adjusted $p$ values. DGE was also performed between Cluster 9 cells and the rest of the matrix to identify upregulated genes in this cluster (Fig. S4).

**Gene set enrichment analysis.** The Python library GSEAPY[101] was used to perform gene set enrichment analysis (GSEA) based on their recommended scRNA-seq workflow to identify the enrichment of gene sets from the *GO Biological Processes 2021* database[102]. GSEA was per-formed on the significant differential genes from Cluster 9 in com-parison to all other cells. Only differentially expressed genes with a log fold change >0 and an adjusted $p$ value < 0.05 were used to identify upregulated gene sets.

**Scoring of dark zone and light zone B cells.** The Scanpy function *score_genes* was used to score enrichment of genes upregulated in LZ B cells in comparison to LZ B cells of the published gene set GSE38696_LIGHT_ZONE_VS_DARK_ZONE_BCELL_UP, which was col-lected from microarrays of gene expression in mouse GC LZ and DZ B cells sorted according to the expression of cell surface molecules CD83 and CXCR4[103].

## BCR analysis

The BCR contigs obtained from the Cell Ranger *vdj* pipeline were aligned to the IMGT mouse reference database using IMGT/High-V Quest[104]. These aligned sequences were further analyzed using a cus-tom Python script to visualize differences in CDRH3 length, identity, and heavy chain isotype. VDJ sequences were clustered into clonotypes using Scirpy[105], a Python package for immune cell receptor repertoire analysis. Clonal clusters were defined using the *define_clonoty-pe_clusters* function using Levenshtein distance based on CDRH3 amino acid sequences with a resolution of 0.5 and matching V genes.

## Testing of the protective efficacy of ANDV mRNA constructs in golden Syrian hamster model

Golden Syrian hamsters, 10-week-old females (Charles River), five animals per group, were immunized twice intramuscularly (gastro-cnemius muscle) with 5 or 25 μg of LNP-encapsulated vaccines, either U-mRNA or m1Ψ-mRNA, with an interval of 21 days. Twenty-one days after the second immunization, the animals were transferred to the BSL-4 facility of the Galveston National Laboratory and challenged intramuscularly with 250 PFU of ANDV. The animals were observed until 28 days after the challenge. If clinical signs were detected, ham-sters were observed more frequently and euthanized based on disease scores. Blood serum was collected from hamsters on days 0 (baseline), 21 (pre-boost), 41 (pre-challenge), 56 (14 dpc), and at euthanasia (terminal). Liver and lung tissue samples were collected at necropsy.

## ELISA for anti-Gn/Gc and anti-N IgG

Reactions were performed as described elsewhere[22] using Falcon™ 96-Well Non-Treated Flat-Bottom Microplates (Corning). Recombinant ANDV Gn$^H$/Gc antigen[8] and recombinant Puumala virus N antigen[77]

which was demonstrated previously to cross-react with ANDV N protein[24] were used as antigens to cover the microplates. Serum samples were tested in twofold dilutions starting from 1:100, in duplicates. The secondary antibody was HRP-conjugated goat anti-hamster IgG (H + L) Cross Adsorbed (Invitrogen) in dilution 1:1500 as determined by checkerboard titration. After incubation with 1-Step Ultra TMB-ELISA (Thermo Fisher Scientific) in the dark for 10–15 min, the reactions were terminated by the addition of 1 N sulfuric acid and detected on Synergy HT Microplate Reader (Bio Tek) at 450 nm. A pool of ten serum samples from hamsters never exposed to hantaviruses was used as a negative control, whereas a pool of five serum samples from hamsters that survived the challenge with ANDV in previous experiments was used as a positive control.

Mouse serum antibody analysis was done as follows. NUNC Maxisorp plates (Thermo Fisher Scientific) were coated with 2 µg recombinant ANDV Gn$^H$/Gc antigen then blocked with 0.5% BSA in 1× PBS. Mouse serum samples were tested in threefold dilutions starting from 1:100 in duplicates. Anti-mouse IgG (Fc) labeled with HRP (Bethyl Laboratories) secondary antibody was used at a dilution of 1:36,000. TMB substrate (BD Biosciences) was added for 20 min and stopped by the addition of 0.2 N H$_2$SO$_4$. Plates were read on a Synergy HT Microplate Reader (Bio Tek) at 450 and 570 nm.

### Detection of ANDV- and SNV-neutralizing antibodies

The plaque reduction neutralization test was performed with hamster sera in 96-well format. Serum samples were inactivated at 56 °C for 30 min, and twofold dilutions in MEM with 10% guinea pig complement (MP Biologicals) were mixed with ~50 PFU of ANDV or SNV, starting from serum dilution 1:10 (final dilution 1:20 after mixing with virus). The mixtures were incubated at 37 °C for 1 h and placed on a monolayer of Vero cells. After 1 h incubation, the inocula were removed, and MEM with 10% FBS and 0.5% methylcellulose overlay was added onto cells. The plates were incubated for 6 days, then fixed in 10% buffered formalin and removed from BSL-4 biocontainment for immunostaining. The immunostaining was performed with a mixture of human monoclonal antibodies as described above, and secondary HRP-conjugated goat anti-human antibody (SeraCare). The reactions were visualized with ImmPACT AEC Substrate (Vector Laboratories). The plates were observed under CKX53 inverted microscope (Olympus) at 10× magnification, and 50% half-maximal inhibitory concentration (IC$_{50}$) values of serum samples were determined.

### Statistical statement

All statistical analyses were performed with GraphPad Prism, version 9.5.0. Non-parametric unpaired Kruskal–Wallis test with Dunn's correction for multiple comparisons, and pairwise comparisons of mean ranks between the variables was used. The differences at $p < 0.05$ were considered statistically significant. The Yates corrected Chi-square and Fisher's exact tests were used for the comparison of survival rates in groups of experimental animals.

### Reporting summary

Further information on research design is available in the Nature Portfolio Reporting Summary linked to this article.

### Data availability

The mRNA vaccine construct sequence has been deposited in GenBank under accession number PP784251. Reference sequence of ANDV segment M used for the development of mRNA is available in GenBank under accession code NC_003467 [https://www.ncbi.nlm.nih.gov/nuccore/NC_003467]; the same virus was used in our in-vitro and in-vivo studies. Sequence of SNV used in the in-vitro studies is available under accession number AF281850. Single-cell sequencing data are available via the Gene Expression Omnibus database (GEO) under accession code GSE240064. Data used for graphs are available in the Source Data file. All other data are available from authors upon request. Source data are provided with this paper.

### Code availability

Scripts for analysis of the single-cell RNA-seq and BCR-seq data are available as Jupyter notebooks at https://github.com/IGlab-VUMC/Andes_Virus_SC-analysis.

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

## Acknowledgements

We thank Dr. Jay Hooper (USAMRIID, Frederick, MD, USA) for providing the recombinant Puumala virus nucleoprotein antigen used for N-IgG ELISA. We thank Meredith Weglarz at the UTMB flow cytometry and cell sorting core laboratory for expert assistance with cell sorting. This project was funded by UTMB intramural funds. A.B. was supported by grants R01 AI41661-01 and P01 AI150585, and R.K.A. was supported by grants 5DP2AI154410 and 5R00AI14576.

## Author contributions

I.V.K. participated in the vaccine design, production, in-vitro evaluation, protection experiment in hamsters, and data analysis; R.S.A. participated in the vaccine production, in-vitro evaluation, and single-cell BCR sequencing; L.P., K.K., K.A.G., K.R.H., K.H., N.G.W., and R.K.A. participated in the analysis of the GC development and antibody responses in mice; R.K.A. supervised the GC analysis; P.T.W. and I.S.G. performed BCR sequence and scRNA-seq analysis; C.M. supervised hamster challenge and sampling at BSL-4 environment; T.B.E. and J.E.C. developed and produced ANDV antigens and antibodies; W.J.M. performed LNP encapsulation of mRNA; V.P. performed electron microscopy; M.A.G.-B. provided expertise on RNA biology; A.B. conceived the study, provided the general supervision and conceptualization, and participated in data analysis. All co-authors participated in the writing and editing, and in the preparation of figures.

## Competing interests

M.A.G.-B. and A.B. declare that they have equity in Emervax, Inc., a company developing and commercializing RNA-based vaccines.
