## [Peer Review File · Nature Communications]

Andes virus mRNA vaccines: comparison of unmodified and modified mRNA platforms for B cell responses and protectionReviewers' Comments:

Reviewer #1:

Remarks to the Author:

The manuscript by Kuzmin et al. "Andes virus mRNA vaccines: comparison of unmodified and modified mRNA platforms" reports immunogenicity and efficacy of novel mRNA vaccines for hantavirus pulmonary syndrome in humans, an emerging infectious disease with high fatality rates. The manuscript is relevant as it compares in a head-to-head approach two distinct mRNA platforms in various rodent models with a focus on B cell responses. It is also timely as it reports generation of first ANDV mRNA vaccine candidates and their efficacy. The authors report an unexpected comparable immunogenicity of the unmodified and modified mRNA vaccine with respect to GC reaction, class-switching and SHM in mice. These vaccines also induce comparable binding and neutralizing antibodies against ANDV and both confer full protection in the hamster challenge model. These findings are highly relevant for vaccinology. Overall, the data are clearly presented and discussed. The methods are generally clearly described. There are several areas in which this manuscript could be improved and these are outlined below.

General comments:

1. The significance of the results presented is limited due to the high vaccine doses used in all models throughout the study (B6 5ug; Balb/c 25 ug; hamster 25 ug). The authors discuss that lower doses may result in variability in GC reaction and antibody profiles. It is necessary to experimentally address this assumption. This is important for understanding of the immunogenicity profiles of the two mRNA platforms and would rule out potential masking effects due to antigen overload. Lower doses benefit also of translatability value.
2. The GC reaction has been evaluated after the first dose. Since this is an ongoing process, it may differ upon delivery of the second dose and diverge between unmodified and modified mRNA vaccines. This limitation should be at least discussed. Furthermore, while certain reasoning for utilization of Balb/c for the scRNA-seq analysis has been provided, it remains unclear whether frequencies of GC B cells in the various mouse models are comparable. Having scRNA-seq data at hand the authors could perform a compositional analysis, using e.g. scCODA for credible changes in cell type composition (Buttner et al., Nat Commun 2021), to confirm comparable response patterns in the two mouse strains. Using same methodology, variability in populations of B cells between tested vaccine platforms can be evaluated, too. Experiments with B6 and Balb/c using an overlapping dose and measuring titers could also be informative.
3. Considering the multispecies approach the authors should discuss apparent discrepancies between mouse and hamster data. Specifically, if GC reaction is comparable for the two mRNA vaccines (in mice) after the first dose why are Ig titers negligible in hamster immunized with the nucleoside-modified mRNA construct? Is this a potential species-specific difference? Ig titers should be evaluated for mice, too.

Specific comments:

Figure 1: An explanation for performing the Gc/Gc measurements and gene expression with the A549 cell line, a type II pneumonocyte, is necessary. Considering the intended i.m. and not mucosal delivery, analyzing antigen presenting cells rather than epithelia seems more informative.

Panel D indicates transcripts detected in U-mRNA but not in Ψ -mRNA transfected cells. However, Table S1 indicates ND for most IFN α members, please clarify.

What are the three electron micrographs presented in panel E? What vaccine preparation(s)?

Figure 2: Details for gating should be provided, i.e. what markers are on the dump channel? It is unclear whether a post-hoc test was used for the Kruskal-Wallis analysis (see also comments regarding statistical analysis).

Figure 3: Information of used vaccine dose should be provided in the figure or in the legend. Details on analyzed cells should be provided, e.g. numbers of sequenced cells, frequencies of low

quality and excluded cells.

Investigations should be expanded by integrating to compositional analysis, please see suggestions above.

Panel 2B seems duplicated in S1 (S1B), please amend.

Material and methods: there is not sufficient description for the statistical analysis. It should be indicated whether data have been tested for normal distribution prior to selection of the statistic test. If not normally distributed, which is very likely with few data points, non-parametric statistics should be applied.

Results and discussion: Benefits of mRNA modifications for antibody responses and GC have been tested only in rodents and thus statements should be specific and carefully amended throughout the manuscript to maintain accuracy.

Whether T cell responses are affected by mRNA modification (U-mRNA vs. Ψ -mRNA) has not been addressed in this manuscript. Accordingly, the focus of the work should be clear from the very beginning, already in the title (narrow to B cell responses). Potential effects on T cell responses could be mentioned in the discussion part.

Reviewer #2:

Remarks to the Author:

The manuscript by Kuzmin and others outlines a series of experiments aimed at comparing a modified and unmodified mRNA vaccine platform against Andes hantavirus infection and disease. As the authors point out, previous to this work numerous groups have published on protective vaccines against hantaviruses, including cross-protection of vaccines based on distantly related hantaviruses, in the same lethal Andes virus challenge model used here. Further, comparisons of modified and unmodified mRNA vaccines have been conducted with other human pathogens both in vitro and in vivo. Overall, this paper is well written and the experiments seem appropriate, however the uniqueness of the work is minimal. I would highly encourage the authors to develop similar mRNA vaccines against at least one other HPS or better HFRS causing hantavirus and determine if cross protection with this platform is achievable in the lethal Andes hamster model.

Minor comments:

1. Page 6 (results): Please specific most recognize NC_003467 as ANDV 9717869. Perhaps add this identification here as well.

2. Dosing on Pages 9-10: The authors list single dose of 5 ug of recombinant antigen on page 9 and 20 ug of recombinant antigen on page 10. Is this a typo or were the dosing different (if different, perhaps explain why)? How do these doses compare to those used in humans?

3. Page 12: The authors state that further studies are needed to determine "the significance of the persistence of vaccine RNA following vaccination". Would persistence of antigen not be equally or perhaps more important to determine?

4. Page 16 (discussion): human to human transmission of ANDV is proven but I do not believe airborne transmission has been definitively established.

5. Page 17: The authors state that "...both constructs induced equal formation of germinal centers (Fig,2)..." yet they did not look at germinal centers but rather B cells with germinal center markers. This should be rephrased. Are there tissue sections that can be analyzed microscopically?

6. Page 17-18: "... except the presence of the cluster 9 with upregulation of genes and enrichment pathways involved in the innate immune response..." What about at lower doses? There seems to be less of a difference.

7. Page 27: Detection of ANDV- and SNV-neutralization antibodies, plaque reduction or focus forming reduction?

Reviewer #3:

Remarks to the Author:

In this study the authors compare the immune responses to modified and unmodified mRNA vaccines encoding the Andes virus glycoproteins Gn and Gc in mice and Syrian hamsters. In the mouse system the authors quantify germinal center (GC) B cells after a single dose immunization by flow cytometry, compare their transcriptomes by scRNA-seq and their hypermutation and switching status by scVDJ-seq. They do not find any obvious differences in immune responses to modified and unmodified mRNA vaccines with the exception of a cluster of GC B cells with type I IFN signaling signature unique to the unmodified mRNA vaccine. In hamsters they analyze the antibody titers, their neutralization capacity after one and two doses of vaccination as well as the protection against infection after the second dose. While two doses of both vaccines conferred protection, a single dose of unmodified but not modified vaccine was sufficient to induce detectable level of antibodies. The manuscript is clearly written, and the experiments are overall sound. However, the study is overall descriptive, does not bring any conceptual advances, and from a practical perspective it is not guaranteed that the main findings will hold true in humans. It therefore may be more suitable for a specialized journal.

Specific points:

- Can the authors explain a seeming discrepancy between the identical magnitude of GC reaction after a single dose of modified and unmodified mRNA vaccines in mice and a rather drastic difference in antibody titers in hamsters (Fig. 5B), especially after the first dose? Is that explained by a difference between the two species or a difference in used readouts? How do antibody titers look in mice?
- The detection of vaccine mRNA in GC B cells is potentially interesting as it could have major consequences for the GC reaction (for example if B cells can "make their own antigen" that could decrease the stringency of competition and affect affinity maturation). It would be interesting to test if GC B cells translate the vaccine.
- The finding that both versions of the vaccine result in an overall comparable response could for example mean that the unmodified vaccine on one hand enhances innate immune responses and on the other may inhibit its own translation thus "pressing accelerator and brake at the same time". This is a testable hypothesis, however neither the parameters of the in vivo innate immune activation nor the in vivo translation of the vaccine was assessed in this study.
- The value of the scRNA/VDJ-seq experiment in this study is unclear. It is also unclear why to include unpurified lymphocytes from a control mouse in a scRNA-seq analysis of GC B cells. Moreover, it could make sense to provide some annotation of GC B cell clusters (e.g DZ vs LZ) and quantify the distribution of cells from mice with different vaccination strategies across all clusters. Some analysis of clonal composition using the scVDJ-seq data could be added. How polyclonal is the response? Are there public clones? Is there a preferential usage of particular V segments? etc.

Reviewer #1 (Remarks to the Author):

The manuscript by Kuzmin et al. “Andes virus mRNA vaccines: comparison of unmodified and modified mRNA platforms” reports immunogenicity and efficacy of novel mRNA vaccines for hantavirus pulmonary syndrome in humans, an emerging infectious disease with high fatality rates. The manuscript is relevant as it compares in a head-to-head approach two distinct mRNA platforms in various rodent models with a focus on B cell responses. It is also timely as it reports generation of first ANDV mRNA vaccine candidates and their efficacy. The authors report an unexpected comparable immunogenicity of the unmodified and modified mRNA vaccine with respect to GC reaction, class-switching and SHM in mice. These vaccines also induce comparable binding and neutralizing antibodies against ANDV and both confer full protection in the hamster challenge model. These findings are highly relevant for vaccinology. Overall, the data are clearly presented and discussed. The methods are generally clearly described. There are several areas in which this manuscript could be improved and these are outlined below.

Response: We thank the Reviewer for the comments that the “the manuscript is relevant”, “is timely”, the results are “unexpected”, “the data are clearly presented and discussed”, and “the methods are generally clearly described”.

General comments:

1. The significance of the results presented is limited due to the high vaccine doses used in all models throughout the study (B6 5ug; Balb/c 25 ug; hamster 25 ug). The authors discuss that lower doses may result in variability in GC reaction and antibody profiles. It is necessary to experimentally address this assumption. This is important for understanding of the immunogenicity profiles of the two mRNA platforms and would rule out potential masking effects due to antigen overload. Lower doses benefit also of translatability value.

Response: We performed a new protective efficacy study in hamsters in which we tested a 5 µg vaccine dose. The data on the immune response and the protection are included in Fig. 5 and Fig. 6, respectively. We also performed immunogenicity studies in two strains of mice (C57BL/6 and BALB/c) with various doses, including a very low dose of 1.5 µg, to compare germinal center responses (Fig. 2C–I), antibody responses (Fig. S3) and follicular helper T cell responses (Fig. S2). These data demonstrated modest or no beneficial effect of RNA modification on antibody responses in hamsters, B cell responses and mice, and IgG response in mice.

2. The GC reaction has been evaluated after the first dose. Since this is an ongoing process, it may differ upon delivery of the second dose and diverge between unmodified and modified mRNA vaccines. This limitation should be at least discussed.

Response: We have added the following comment: “One limitation of the study was that we evaluated GC responses only following the priming dose. Since GCs can last many months⁹⁰, it would be interesting in future studies to assess how each of these mRNA vaccine platforms drive GC durability, and SHMs longitudinally including following boost” (page 22 paragraph 2).

Furthermore, while certain reasoning for utilization of Balb/c for the scRNA-seq analysis has been provided, it remains unclear whether frequencies of GC B cells in the various mouse models are comparable.

Response: To address this point we conducted two additional head-to-head experiments in C57BL/6J and BALB/c mice with the same batches of modified or non-modified mRNA vaccines (Fig. 2C,D, S1, S2C–F). These extensive new studies demonstrated that germinal center B cell responses and antibody responses in BALB/c mice are comparable or slightly greater compared to that in C57BL/6J mice.

Having scRNA-seq data at hand the authors could perform a compositional analysis, using e.g. scCODA for credible changes in cell type composition (Buttner et al., Nat Commun 2021), to confirm comparable response patterns in the two mouse strains. Using same methodology, variability in populations of B cells between tested vaccine platforms can be evaluated, too.

Response: As commented above, the flow cytometry data on germinal center responses in the two mouse strains demonstrated similar profiles, and therefore we believe a new resource intensive single-cell sequencing experiment was not required. However, we have now performed scCODA compositional analysis to compare the two vaccine platforms, which is shown in Fig. 3C, and referred to the Buttner et al., Nat. Commun. 2021 in

Results and in Methods. The U-mRNA and m1Ψ-mRNA groups showed comparable composition of cell counts by cluster, with the only quantifiable difference being the presence of Cluster 9 cells for U-mRNA treated mice.

Experiments with B6 and Balb/c using an overlapping dose and measuring titers could also be informative.

Response: As we indicate above, we conducted the mouse strain specific germinal center responses head-to-head with the same batches of mRNA vaccines and saw no difference or modestly higher responses in BALB/c mice (Fig. 2C-D, S1, S2C-F). Additionally, we assessed GnGc specific IgG and IgM titers in both strains of mice (5 μg dose), as well at a very low dose of 1.5 μg on day 12 post immunization in C57BL/6 mice (Fig. S3). Both IgM and IgG titers were very low at this time point, with IgM titers being identical between vaccine constructs in all conditions and IgG titers being moderately higher in the modified mRNA vaccine group.

3. Considering the multispecies approach the authors should discuss apparent discrepancies between mouse and hamster data. Specifically, if GC reaction is comparable for the two mRNA vaccines (in mice) after the first dose why are Ig titers negligible in hamster immunized with the nucleoside-modified mRNA construct? Is this a potential species-specific difference? Ig titers should be evaluated for mice, too.

Response: As discussed above, we conducted additional experiments and quantified Ig titers for the two vaccine platforms at two vaccine doses in two mouse strains (Fig. S3). This analysis revealed the titers post primary immunization is low, which is in agreement with the low IgG titers in hamsters after a single dose on day 21 (Fig. 5B). While we cannot directly state why the Ig titers are negligible after the primary immunization, we speculate that there is a possible immunodominance of cryptic epitopes for the early extrafollicular B cell response which is not detected by the recombinant GnGc protein. As the germinal center takes about week to form, the serum antibody response that is derived specifically from the germinal center reaction takes several weeks to reach appreciable levels (Schiepers et al *Nature* 2023). Due to this, we posit that the detectable binding antibody response specific to protective epitopes is likely germinal center derived as this response takes two immunizations and several weeks to develop. As the kinetics of the germinal center response at the cellular level, and the protective antibody response in the serum are two different parameters, we do not believe there is a discrepancy. Nevertheless, we do note in the manuscript that since we did not evaluate germinal center responses directly in hamsters, there may in fact be species differences. We have updated the Discussion to include these points (page 22 paragraph 2).

Specific comments:

Figure 1: An explanation for performing the Gc/Gc measurements and gene expression with the A549 cell line, a type II pneumocyte, is necessary. Considering the intended i.m. and not mucosal delivery, analyzing antigen presenting cells rather than epithelia seems more informative.

Response: A549 cells is the commonly accepted IFN-I-competent cell line for evaluation of the innate immune responses *in vitro* regardless of the specific application of compounds. To address the Reviewer's comment, we performed new experiments which initially included transfections of 293T and HeLa cells. However, these cells demonstrated negligible innate immune response to either construct which is expected due to their altered transcription profiles. We next made several attempts to transfect primary human monocytes from several donors, but the transfected efficiency was very low. Finally, we were able to address the Reviewer's comment with human monocytic cell line THP1. These experiments demonstrated the cytokine profile which was very similar to that in A549 cells. These data are added as the right panel in Fig. 1D.

Panel D indicates transcripts detected in U-mRNA but not in Ψ-mRNA transfected cells. However, Table S1 indicates ND for most IFNα members, please clarify.

Response: We thank the Reviewer for pointing out at the error, we fixed it.

What are the three electron micrographs presented in panel E? What vaccine preparation(s)?

Response: We have clarified in the figure legend (U-mRNA).

Figure 2: Details for gating should be provided, i.e. what markers are on the dump channel? It is unclear whether a post-hoc test was used for the Kruskal-Wallis analysis (see also comments regarding statistical analysis).

Response: 1. We have now provided details for gating in the new panel B of the figure.

2. We modified the figures (except Fig. 1B *in vitro* data) for the non-parametric Kruskal-Wallis test with Dunn's correction for multiple comparisons, and pairwise comparisons of mean ranks between the variables.

Figure 3: Information of used vaccine dose should be provided in the figure or in the legend.

Response: We have now added it in the legend for additional clarity.

Details on analyzed cells should be provided, e.g. numbers of sequenced cells, frequencies of low quality and excluded cells.

Response: We have included details of the scRNA-seq and VDJ-seq in Suppl. Data 2. This spreadsheet includes metrics output from Cell Ranger for sequencing alignment, along with details on further filtering performed.

Investigations should be expanded by integrating to compositional analysis, please see suggestions above.

Response: We used scCODA from Buttner et al. to analyze differences in cellular subtypes across treatment groups (Fig. 3C, D). This analysis agreed with the previous observation that the only difference between vaccine groups was the enrichment of Cluster 9 cells in the U-mRNA treated mice.

Panel 2B seems duplicated in S1 (S1B), please amend.

Response: The duplicating panel removed, the figure legend and the text amended.

Material and methods: there is not sufficient description for the statistical analysis. It should be indicated whether data have been tested for normal distribution prior to selection of the statistic test. If not normally distributed, which is very likely with few data points, non-parametric statistics should be applied.

Response: We agree with the Reviewer. With the limited datasets the distribution indeed was not normal (and could not be even assessed for that). Although we have no reason to expect that our samples originated from otherwise normally distributed populations, out of caution we replaced all parametric tests with non-parametric which did not change the essence of our findings. All figures and relevant sections are changed accordingly.

Results and discussion: Benefits of mRNA modifications for antibody responses and GC have been tested only in rodents and thus statements should be specific and carefully amended throughout the manuscript to maintain accuracy.

Response: In most cases, the text states explicitly that all comparisons have been performed in rodent models. We have now better emphasized this in the last sentence of the Abstract and the last sentence of Introduction. We also refined our comments on the difference in expression of IL-1 and IL-1 receptor antagonist in human and rodent cells, and difference in biological effects of oligonucleotide synthetase 1 in human and rodent cells (the last paragraph of the Discussion).

Whether T cell responses are affected by mRNA modification (U-mRNA vs. Ψ -mRNA) has not been addressed in this manuscript. Accordingly, the focus of the work should be clear from the very beginning, already in the title (narrow to B cell responses). Potential effects on T cell responses could be mentioned in the discussion part.

Response: We have extended the title to mention B cell responses and added analysis of T follicular helper T cells (Fig. S2). We have also added a statement that CD8 T cell responses were not characterized (the last paragraph of the Discussion).

Reviewer #2 (Remarks to the Author):

The manuscript by Kuzmin and others outlines a series of experiments aimed at comparing a modified and unmodified mRNA vaccine platform against Andes hantavirus infection and disease. As the authors point out, previous to this work numerous groups have published on protective vaccines against hantaviruses, including cross-protection of vaccines based on distantly related hantaviruses, in the same lethal Andes virus challenge model used here. Further, comparisons of modified and unmodified mRNA vaccines have been conducted with other human pathogens both in vitro and in vivo. Overall, this paper is well written and the experiments seem appropriate, however the uniqueness of the work is minimal. I would highly encourage the authors to develop similar mRNA vaccines against at least one other HPS or better HFRS causing hantavirus and determine if cross protection with this platform is achievable in the lethal Andes hamster model.

Response: 1. We thank the Reviewer for the comment that “this paper is well written and the experiments seem appropriate”. The initial comparison by Pardi et al. (PMID: 29739835) which involved the mouse model of

influenza virus did not include analysis of B cell repertoire and antibody repertoire. We are unaware of other studies in which the effect of pseudouridine modification of linear mRNA-based vaccine platform was investigated in detail.

2. The uniqueness of this study includes, but not limited to the two points: (A) This is the first development and testing of effective RNA vaccines against a highly lethal and transmissible virus or against any hantavirus to our knowledge. (B) Demonstration that the effect of pseudouridine modification on the immune response and protection can be non-detectable or modest. While this manuscript was in revision, a study published in *Nature* demonstrated that N1-pseudouridinylation of mRNA causes +1 ribosomal frameshifting (PMID: 38057663). These data, together with our data on modest or undetectable beneficial effects of mRNA modification of the antibody response and protection argue for careful consideration of non-modified mRNA as the vaccine platform. To better emphasize the importance of our study, we have now added this consideration in the last paragraph of the Discussion.

3. Development and testing of a vaccine against another hantavirus would take a significant time and effort and was out of scope of this study; however, we plan to develop additional hantavirus vaccines in a separate project.

Minor comments:

1. Page 6 (results): Please specific most recognize NC_003467 as ANDV 9717869. Perhaps add this identification here as well.

Response: We have added this information in Methods (paragraph 1).

2. Dosing on Pages 9-10: The authors list single dose of 5 ug of recombinant antigen on page 9 and 20 ug of recombinant antigen on page 10. Is this a typo or were the dosing different (if different, perhaps explain why)? How do these doses compare to those used in humans?

Response: We thank the Reviewer for catching this typo. The recombinant antigen dose on page 9 was not indicated, it now is (20 µg). Since the actual quantity of *in vivo* translation of the mRNA vaccines to protein is unknown, it is impossible to exactly dose match a protein vaccine with an mRNA vaccine. As such, we chose a standard middle-range dose of 20 µg to immunize mice as a control. In terms of mRNA vaccine doses, we tested a wide range of doses (1.5 µg to 25 µg) that are well in the dose range of preclinical rodent models for mRNA vaccines. There is no universal agreement on if doses should be scaled from animal models, as many times vaccine doses are quite similar across species. SARS-CoV-2 mRNA vaccines that have been used in humans have ranged between the low µg range in children to 250 µg in adults. While doses are sometimes scaled, this is not generally based on mass, but rather body surface area. One argument against scaling vaccines is that they are largely considered “locally acting drugs” as in they can cause a local reaction at the site of injection and drain to the nearest lymph nodes. This is one reason why there is significant overlap, and often very similar, if not identical, vaccine doses given to mice, rabbits, monkeys, humans, and cattle despite massive size differences.

3. Page 12: The authors state that further studies are needed to determine "the significance of the persistence of vaccine RNA following vaccination". Would persistence of antigen not be equally or perhaps more important to determine?

Response: We performed experiments vaccinating mice with both vaccine constructs and evaluating different populations of cells from draining lymph nodes for expression of the vaccine antigen by flow cytometry on days 1, 3, 6, and 14 post vaccination. Selected representative images are shown in Fig. S7 and S8. The data did not show reproducible presence of the antigen in B cells, but did show some antigen in myeloid cells. However, as the total numbers of the antigen-positive cells were small, we consider them inconclusive and suggest that further studies are required.

4. Page 16 (discussion): human to human transmission of ANDV is proven but I do not believe airborne transmission has been definitively established.

Response: The NEJM paper describing the 2018 – 2019 human outbreak (ref. 49) states that “On the basis of both the epidemiologic and genomic investigations of person-to-person transmission events, it appears that inhalation of droplets or aerosolized virions may have been the routes of infection.” We have now softened the statement as follows: “Hantaviruses pose a significant public health threat in parts of the world, and their pandemic potential was highlighted by human-to-human transmission of ANDV in 2019 - 2020 likely by inhalation of droplets or aerosolized virions” (the 1st sentence of the Discussion).

5. Page 17: The authors state that “..both constructs induced equal formation of germinal centers (Fig,2)..” yet they did not look at germinal centers but rather B cells with germinal center markers. This should be rephrased. Are there tissue sections that can be analyzed microscopically?

Response: To address this point we have now conducted extensive histological analysis of hundreds of germinal centers in both inguinal and popliteal lymph nodes in mice immunized with either non-modified or modified mRNA vaccines. This data is now shown in Fig. 2E-I and is in agreement with the flow cytometric analysis of germinal center B cells showing that the two vaccine induce comparable responses.

6. Page 17-18: “... except the presence of the cluster 9 with upregulation of genes and enrichment pathways involved in the innate immune response...” What about at lower doses? There seems to be less of a difference.

Response: We performed new mouse experiments (Fig. S1, S2, S3), in which we assessed if lower doses would elicit a different result than the 25 µg dose used in the single-cell sequencing experiment. While the magnitude of the germinal center response was lower as expected, modification of mRNA resulted in no or limited increase in the response at both the low (5 µg) and the very low (1.5 µg) doses. The higher dose was selected as mouse lymph nodes are extremely small, and with a lower vaccine dose we may not be able to collect immune cells in the amounts sufficient for single-cell sequencing.

7. Page 27: Detection of ANDV- and SNV-neutralization antibodies, plaque reduction or focus forming reduction?

Response: We consider that FRNT is a variant of PRNT and as such used the most basic and commonly known term PRNT.

Reviewer #3 (Remarks to the Author):

In this study the authors compare the immune responses to modified and unmodified mRNA vaccines encoding the Andes virus glycoproteins Gn and Gc in mice and Syrian hamsters. In the mouse system the authors quantify germinal center (GC) B cells after a single dose immunization by flow cytometry, compare their transcriptomes by scRNA-seq and their hypermutation and switching status by scVDJ-seq. They do not find any obvious differences in immune responses to modified and unmodified mRNA vaccines with the exception of a cluster of GC B cells with type I IFN signaling signature unique to the unmodified mRNA vaccine. In hamsters they analyze the antibody titers, their neutralization capacity after one and two doses of vaccination as well as the protection against infection after the second dose. While two doses of both vaccines conferred protection, a single dose of unmodified but not modified vaccine was sufficient to induce detectable level of antibodies. The manuscript is clearly written, and the experiments are overall sound. However, the study is overall descriptive, does not bring any conceptual advances, and from a practical perspective it is not guaranteed that the main findings will hold true in humans. It therefore may be more suitable for a specialized journal.

Response: 1. We thank the Reviewer for the comment that “the manuscript is clearly written, and the experiments are overall sound.” Regarding the importance of the study, please see our response to comment 1 of the Reviewer 2.

Specific

points:

- Can the authors explain a seeming discrepancy between the identical magnitude of GC reaction after a single dose of modified and unmodified mRNA vaccines in mice and a rather drastic difference in antibody titers in hamsters (Fig. 5B), especially after the first dose? Is that explained by a difference between the two species or a difference in used readouts? How do antibody titers look in mice?

Response: We performed additional experiments on induction of germinal centers in mice, in which we compared two different vaccine doses in two mouse strains (see the data in the new Fig. 2, S1, S2 and the responses to comments of Reviewer 1). We also performed an additional study with hamsters in which we tested a lower (5 µg) dose of each U-mRNA and m1Ψ-mRNA (Fig. 5, 6). The serological assays of the hamster immune sera were repeated twice, which resulted in the same results. As the serological methods were similar for hamsters and mice, the difference is explained by biological differences between the two animal models. Our comments on the new mouse data are included in our responses to the comments of the Reviewer 1.

- The detection of vaccine mRNA in GC B cells is potentially interesting as it could have major consequences for

the GC reaction (for example if B cells can “make their own antigen” that could decrease the stringency of competition and affect affinity maturation). It would be interesting to test if GC B cells translate the vaccine.

Response: We performed additional experiments in which we attempted to detect the antigen in immune cells by flow cytometry (Fig. S7 and S8) – please see our response to comment 3 of the Reviewer 2.

- The finding that both versions of the vaccine result in an overall comparable response could for example mean that the unmodified vaccine on one hand enhances innate immune responses and on the other may inhibit its own translation thus “pressing accelerator and brake at the same time”. This is a testable hypothesis, however neither the parameters of the in vivo innate immune activation nor the in vivo translation of the vaccine was assessed in this study.

Response: We appreciate this point from the Reviewer and have raised this point in the discussion. To attempt to address and expand upon this point, we performed additional experiments in which we vaccinated mice with both vaccines and determined cytokine responses in their muscles and draining lymph nodes by qRT-PCR on days 1 and 3 after vaccination. The level of cytokines was equal between the two vaccines and equal to the mock-vaccinated group (Table S2). In other new experiments, we quantified the ANDV antigen in immune cells isolated from mice (Fig. S7, S8), but the results appeared to be inconclusive (see our response to Comment 3 of the Reviewer 2).

- The value of the scRNA/VDJ-seq experiment in this study is unclear. It is also unclear why to include unpurified lymphocytes from a control mouse in a scRNA-seq analysis of GC B cells. Moreover, it could make sense to provide some annotation of GC B cell clusters (e.g DZ vs LZ) and quantify the distribution of cells from mice with different vaccination strategies across all clusters. Some analysis of clonal composition using the scVDJ-seq data could be added. How polyclonal is the response? Are there public clones? Is there a preferential usage of particular V segments? etc.

Response: 1. Regarding the value of the single-cell VDJ-seq, while our *in vitro* gene expression data showed no or reduced upregulation of innate immune response genes associated with mRNA modification (Fig. 1D, Table S1), in concordance with previous studies, further *in vivo* evaluation of the cytokine responses showed no difference between the two vaccine constructs (Table S2). We performed scRNA-seq to determine if this effect is translated in populations of germinal center B cells. The single-cell VDJ-seq was performed in order to determine whether there were differences in the BCR characteristics of germinal center B cells between the groups, with a focus on class-switched BCRs. This includes levels of SHM and proportions of isotypes expressed, and is enhanced by the additional analyses of clonality, public clones, and V gene usage as suggested.

2. We presume that the Reviewer comments the control group of mice which were mock-vaccinated with empty LNPs. This group was included to better discriminate the changes associated with the different RNA vaccine platforms, as opposed to vaccination with any mRNA vaccine platform.

3. We have now performed scoring of cells for DZ/LZ gene expression (please see the new Fig. 3G). However, the DZ/LZ distinction is not clear as the transcriptional states of the GC B cells represent a spectrum of DZ/LZ-like gene expression. Also, these DZ/LZ relevant genes are largely evenly distributed across the clusters, suggesting that they are not critical for defining the observed cellular states. Flow cytometric staining also revealed equal distribution of DZ and LZ clustering regardless of vaccine dose or vaccine used (Fig S1). There was a moderate increase in LZ phenotype cells in BALB/c mice when compared to C57B6/J mice (Fig S1).

4. We have now performed quantification of the distribution of cells from different groups across clusters using scCODA for compositional analysis. We have also performed more extensive analysis of the VDJ-seq data, including analysis of clonality, clonal clustering to identify public clones between treatment groups, and usage of heavy and light variable genes. These analyses demonstrated that the polyclonal expansion and variable gene usage is comparable between the U-mRNA and m1Ψ-mRNA cells (Fig. 4D,F). We did find an enrichment of public clones (based on matching variable genes and $\geq 70\%$ CDRH3 identity) between the mRNA and m1Ψ-mRNA groups.

Reviewers' Comments:

Reviewer #1:

Remarks to the Author:

Comments have been adequately addressed.

The title of the figure 3 should include "lymph node B cells" to maintain scientific accuracy. The analysis was restricted to this population.

Reviewer #2:

Remarks to the Author:

The authors have addressed my comments and concerns appropriately.

Reviewer #3:

Remarks to the Author:

In my opinion the revised version of the study remains descriptive, does not bring any conceptual advances, and from a practical perspective it is not guaranteed that the main findings will hold true in humans. In addition, new results with lower vaccine dose reveal a rather dramatic difference in the magnitude of GC response with modified RNA vaccine resulting in a multiple fold increase in GC B cells and Tfh cells (Fig. 2C, D; Fig. S2B, F) as well as increased antibody titers (Fig. S3). This undermines one of the main conclusions of the paper on the "lack of a substantial effect of m1 Ψ mRNA modification on immunogenicity". In my opinion this study is not suitable for publication in Nature Communications.